# CryoEM structures reveal how the bacterial flagellum rotates and switches direction

Prashant K. Singh [ORCID][1], Pankaj Sharma [ORCID][1], Oshri Afanzar[2], Margo H. Goldfarb[1], Elena Maklashina[3,4], Michael Eisenbach [ORCID][5], Gary Cecchini [ORCID][3,4] & T. M. Iverson [ORCID][1,6,7,8] ✉

Bacterial chemotaxis requires bidirectional flagellar rotation at different rates. Rotation is driven by a flagellar motor, which is a supercomplex containing multiple rings. Architectural uncertainty regarding the cytoplasmic C-ring, or 'switch', limits our understanding of how the motor transmits torque and direction to the flagellar rod. Here we report cryogenic electron microscopy structures for *Salmonella enterica* serovar *typhimurium* inner membrane MS-ring and C-ring in a counterclockwise pose (4.0 Å) and isolated C-ring in a clockwise pose alone (4.6 Å) and bound to a regulator (5.9 Å). Conformational differences between rotational poses include a 180° shift in FliF/FliG domains that rotates the outward-facing MotA/B binding site to inward facing. The regulator has specificity for the clockwise pose by bridging elements unique to this conformation. We used these structures to propose how the switch reverses rotation and transmits torque to the flagellum, which advances the understanding of bacterial chemotaxis and bidirectional motor rotation.

The biased random walk of chemotaxis is essential for bacterial survival and pathogenesis[1–3]. This process relies on bidirectional flagellar rotation[1–3] (Fig. 1a) by a motor composed of four rings. The C-ring (C = cytoplasmic), which contains multiple copies of protein subunits called FliG, FliM and FliN, switches the rotation of the flagellum between counterclockwise (CCW) and clockwise (CW). For this reason, it is termed the 'switch'. CCW rotation allows bacteria such as *Escherichia coli* and *Salmonella enterica* to swim straight. Conversely, CW rotation induces tumbling and reorientation with a new trajectory[1–3]. The switch is also the site of torque generation for the flagellum, through an electrostatic interaction with a stator called MotA/B[4]. The switch also connects to the MS-ring (MS = membrane–supramembrane), which transmits both the direction and the speed of rotation to the flagellar rod. Finally, the P- (P = peptidoglycan) and L-rings (L = lipid), which are the bushings of the motor, surround this rod to support and buffer the rotation. High-resolution structures of the rod, export apparatus,

MS-ring, P-ring, L-ring, flagellar hook and flagellar filament have provided insight into the function of these flagellar components[5–13]. However, past structures for the C-ring are at low resolution (for example, refs. [14,15]).

Response regulators affect flagellar rotation and speed. The best studied is the excitatory response regulator CheY (for chemotaxis)[16,17], which biases the flagellum toward CW rotation. Others include the fumarate-sensing quinol:fumarate reductase[18–20], the spermidine-sensing SpeE[21] and the cyclic-di-GMP-sensing YcgR[22,23]. The CW pose[18] may also support switch assembly, as well as assembly and disassembly of the entire flagellum[24,25].

Key unknowns in chemotaxis are how the motor drives both CCW and CW rotation, how response regulators affect rotation and how torque transfers from the stator to the flagellum. To help inform on these controversies, we determined the structures of the *S. enterica* serovar *typhimurium* combined MS- and C-rings in the CCW pose, the

[1]Department of Pharmacology, Vanderbilt University, Nashville, TN, USA. [2]Department of Microbiology & Immunology, Stanford University School of Medicine, Stanford, CA, USA. [3]Molecular Biology Division, San Francisco VA Health Care System, San Francisco, CA, USA. [4]Department of Biochemistry & Biophysics, University of California, San Francisco, CA, USA. [5]Department of Biomolecular Sciences, The Weizmann Institute of Science, Rehovot, Israel. [6]Department of Biochemistry, Vanderbilt University, Nashville, TN, USA. [7]Center for Structural Biology, Vanderbilt University, Nashville, TN, USA. [8]Vanderbilt Institute of Chemical Biology, Vanderbilt University, Nashville, TN, USA. ✉e-mail: tina.iverson@vanderbilt.edu

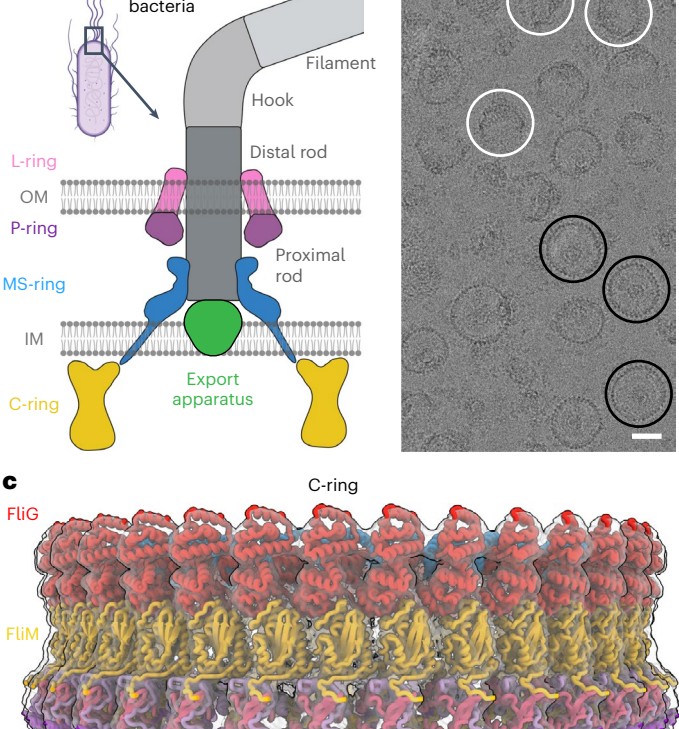

**Fig. 1 | The flagellar motor and structure of the switch. a**, Schematic diagram of the flagellar motor showing the L-ring, P-ring, MS-ring and C-ring, as well as the flagellar rod, hook and filament. The switch is housed within the C-ring and is composed of the FliG, FliM and FliN subunits. OM, outer membrane; IM, inner membrane. **b**, Cropped view of a representative cryoEM micrograph for wild-type MS- and C-rings (1 of 34,381 micrographs) showing the quality of particles used in structure determination. Most particles contain both the MS- and C-rings, although a small number of isolated MS-rings are present. En face views (three are highlighted with black circles) and side views (three are highlighted with white circles) are observed. Tilted views are also observed and give the appearance of a smaller diameter in some cases. Scale bar, 200 Å. The uncropped micrograph is available in the Source data file. Raw micrographs for all structures have been deposited with EMPIAR[76] (https://www.ebi.ac.uk/empiar/) and accession codes EMPIAR-11597, EMPIAR-11891 and EMPIAR-11892. **c**, Surface representation of the C-ring density maps in the CCW pose superimposed on the final model. FliF subunits are shown in blue, FliG subunits are shown in red, FliM subunits are shown in yellow and FliN subunits are shown in shades of pink and purple.

C-ring in the CW pose and the CW pose bound to a response regulator. This complements concomitant work from the Lea group showing the isolated C-ring in the CCW pose, the CW pose and the MotA/B interaction with the C-terminus of FliG[26].

## Results

### Structure of the wild-type C-ring

We formed particles from coexpressed FliF, FliG, a region of FliL, FliM, FliN and FliO[27]. We purified the resultant 6 MDa supercomplex containing both the MS-ring and the C-ring, collected cryogenic electron microscopy (cryoEM) data and determined the structure (Fig. 1b,c, Extended Data Fig. 1a,b, Supplementary Table 1 and Supplementary Video 1). Standard cryoEM workflows could not improve the resolution beyond ~8 Å. We, therefore, used particle subtraction at the level of the micrograph (Extended Data Fig. 1a). This technique improves alignment by obscuring unwanted features in the primary dataset[28], with common applications including removing nanodiscs or detergent

micelles from membrane protein particles. We removed the MS-ring and determined the structure of the C-ring, where the predominant species was a 34-mer (~50% of the particles). Local resolutions ranged from 2.9 to 6.6 Å, and the average resolution was 4.0 Å (Extended Data Fig. 2a and Supplementary Video 1). Other symmetries included a 33-mer (4.5 Å resolution), 35-mer (4.5 Å resolution) and 36-mer (6.7 Å resolution).

We interpreted the 34-mer maps by docking AlphaFold (v.2.0)[29] models of isolated *S. enterica* domains and manually connecting them (Fig. 2a–f, Extended Data Fig. 2b,c and Supplementary Video 2). This identified that the switch subunits loosely organize into layers. Beginning at the side of the C-ring that faces the MS-ring and the membrane, the layers contain $FliF_{514–560}$/$FliG_{1–331}$ in two layers at the top, $FliM_{52–237}$ ($FliM_{mid}$) in the middle, and $FliM_{257–330}$ ($FliM_C$)/$FliN_{45/59/63–137}$ in a 3:1 ratio at the bottom (Fig. 2e).

The density (Extended Data Fig. 3a–o) was consistent with Alpha-Fold models and X-ray crystal structures of isolated domains of FliG[14,30–38], FliM[21,32,37,39–41] and FliN[39,42] from homologues (Extended Data Fig. 2c). However, crystal structures of multi-domain FliG[31,35,38] required substantial interdomain adjustment to match the cryoEM structure (Extended Data Fig. 4a–f). In addition, past work supports FliG as a three-domain protein[38]; however, FliG contains five domains when it is assembled into the switch (Fig. 2b). Therefore, the FliG domains are redefined here as $FliG_{D1}$–$FliG_{D5}$ ($FliG_{1–67}$, $FliG_{73–99}$, $FliG_{107–186}$, $FliG_{196–233}$ and $FliG_{243–331}$; Supplementary Table 2). Interdomain linkers are termed $FliG_{L1}$–$FliG_{L5}$.

Mutagenesis is commonly used to validate cryoEM structures. Given the extensive number of published mutants, designing new mutations was unnecessary. From this, we identified many assembly-deficient mutations[30,32,43] that affect residues that form strong interactions at subunit interfaces (Extended Data Fig. 4g), which explains their impact on switch assembly.

### The CCW pose of the switch

Purified wild-type C-rings exclusively rotate CCW under physiological conditions[44], assigning this as the CCW pose. Furthermore, this structure concurs with the CCW pose shown by tomography[45,46] (Extended Data Fig. 4h). In this structure (Fig. 2a–f), individual FliG subunits form a V shape. When assembled into a 34-mer, the upper regions form inner and outer rings separated by a 30 Å cleft. The inner ring contains $FliF_C$, $FliG_{D1}$ and $FliG_{D2}$, and the outer ring contains $FliG_{D5}$. At a more detailed level, $FliF_C$ forms a curved helix that extends radially from the MS-ring on the membrane side (Fig. 2a). This $FliF_C$ interacts intimately with $FliG_{D1}$ (Fig. 2b) and has strong density for all C-terminal residues of $FliF_C$ (Extended Data Fig. 3a). As $FliG_{D1}$ extends into $FliG_{D2}$, it forms an armadillo motif, which is an α-helical hairpin that permits rotations. This armadillo motif of $FliG_{D2}$ completes the fold of the next $FliG_{D1}$ to form an intercalated structure (Fig. 2g). By contrast, $FliG_{D5}$ of the outer ring distinctly separates from neighbouring subunits (Fig. 2h).

Below these upper rings, armadillo motifs of $FliG_{D3}$ and $FliG_{D4}$ (Fig. 2b,f) intercalate around the ring via domain swaps, as proposed by ref. 38. $FliG_{D4}$ also forms the base of the cavity between the upper rings, and $FliG_{D3}$ binds to FliM (Fig. 2f). The $FliG_{L1}$–$FliG_{L4}$ linkers between these domains have strong density suggesting rigidity (Extended Data Fig. 3b,c,e,f). $FliG_{L3}$ is noteworthy because it both stabilizes the position of $FliG_{D2}$ and makes intimate interactions with a Pro-Ala-Ala (PAA) sequence motif ($FliG_{169–171}$) in an adjacent $FliG_{D3}$ protomer (Fig. 2b,i). Deletion of this PAA motif results in flagellar motors that predominantly rotate CW[35,47].

$FliM_{mid}$ (Fig. 2c,d,f) and its interface with $FliG_{D3}$ resemble crystal structures[21,32,37,39] except for a new helix containing $FliM_{231–256}$ ($FliM_{L2}$). This helix domain swaps with the adjacent subunit and extends to the bottom of the structure. Here $FliM_C$ and $FliN_C$ adopt an organization called a SpoA fold[39,42] and form a heterotetrameric building block that resembles a split lock washer (Fig. 2j,k). These $FliM_C$:3$FliN_C$ building blocks pack into a spiral with $FliM_C$ at the lower edge of the C-ring (Fig. 2c,d,j,k) that is consistent with past biochemical and structural studies[38,48].

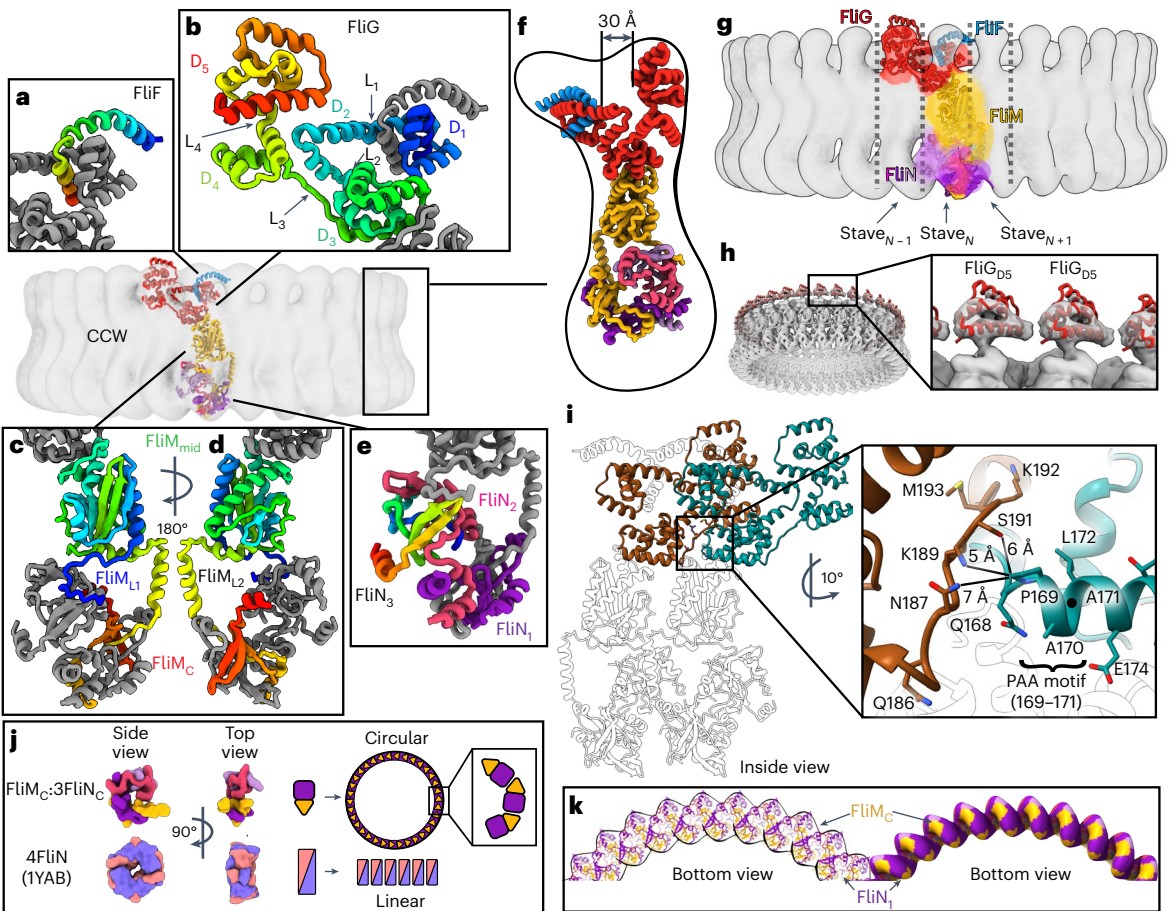

**Fig. 2 | The CCW pose of the switch.** In the global views, FliF is blue, FliG is red, FliM is yellow and FliN is pink and purple. In the insets of **a**–**e**, each of the subunits is coloured from the N-terminus (blue) to C-terminus (red) to highlight the fold. **a**, FliF$_C$ wraps around FliG$_{D1}$. **b**, A FliG protomer folds into five domains: FliG$_{D1}$ (FliG$_{1–67}$), FliG$_{D2}$ (FliG$_{73–99}$), FliG$_{D3}$ (FliG$_{107–186}$), FliG$_{D4}$ (FliG$_{196–233}$) and FliG$_{D5}$ (FliG$_{243–331}$). **c**, The FliM subunit, highlighting FliM$_{L1}$ (FliM$_{31–50}$), FliM$_{mid}$ (FliM$_{51–230}$), FliM$_{L2}$ (FliM$_{231–256}$) and FliM$_C$ (FliM$_{257–330}$). **d**, A 180° rotated view of panel (**c**). **e**, Three FliN$_C$ subunits are similar but non-equivalent. To highlight the fold, only one protomer (FliN$_3$) is coloured from the N-terminus (blue) to C-terminus (red). The remaining two (FliN$_1$ and FliN$_2$) are coloured pink and purple. **f**, A side view of a single FliFGMN unit. An ~30 Å cleft between FliF$_C$–FliG$_{D1/D2}$ and FliG$_{D5}$ is highlighted.

**g**, A single FliFGMN unit participates in three staves. **h**, Density for FliG$_{D5}$ appears to be separated, with the domain having little contact with adjacent subunits. **i**, Interactions between the PAA motif of FliG$_{D3}$ and the adjacent FliG$_{L1}$ linker. **j,k**, Formation of a curved spiral by the FliM$_C$:3FliN$_C$ heterotetramer. **j**, A schematic that compares the open ring of FliM$_C$:3FliN$_C$ in the cryoEM structure to the closed ring of the crystal structure of *T. maritima* FliN$_C$ (1YAB[42]). This comparison highlights that a pure FliN$_C$ superstructure would favour stacked discs in a linear array. FliM$_C$ breaks the symmetry, which is necessary to form the helix along the bottom of the C-ring. **k**, The FliM$_C$:3FliN$_C$ forms a spiral that curves along the base of the C-ring to form a closed circle. A single arc is shown.

## The CW pose of the switch

A distinct pose of the switch supports CW rotation[45,46] and switch assembly[18,19]. To inform on this pose, we determined the 4.6 Å resolution structure of switch particles containing the extreme CW-biased FliG$_{ΔPAA}$ mutation[47]. Symmetry expansion followed by local refinement gave superior results to particle subtraction (Extended Data Fig. 5a). To build the model (Fig. 3a–f), we docked modules from the assembled CCW pose as rigid bodies into the CW maps. Linkers between domains were then built manually.

The CW pose contains significant domain rearrangement, particularly in FliG. Here FliF$_C$–FliG$_{D1}$ and FliG$_{D2}$ of the inner ring and FliG$_{D5}$ of the outer ring each rotate by ~180° (Fig. 3a,b,g–i). As a part of this, FliG$_{D2}$ changes the subunit that it binds, altering the domain swaps (Fig. 3j). These changes have multiple impacts. They reverse the orientation FliG$_{D5}$, which binds the MotA/B stator, and they also reverse FliG$_{D1}$, which binds to the MS-ring. Finally, these rotations increase the size of the cleft between the inner and outer rings from 30 Å to 40 Å (Figs. 2f and 3f). FliG$_{D3}$–FliM$_{mid}$ undergoes smaller positional changes, rotating approximately 25° as a unit with a slight adjustment in the binding interface (Fig. 3c,d,k).

The altered domain swaps in the CW pose (Fig. 3j) may have biological implications. First, the reduced number of swaps could facilitate C-ring assembly[18,19]. The altered domain swaps also suggest how the C-ring supports directional cooperativity[49,50], where a change from CCW to CW in one subunit may trigger a similar change in an adjacent subunit. The likely steric clash in a ring of mixed poses would induce one subunit to change the conformation of the adjacent subunit.

Comparison of the CCW and CW poses (Fig. 3g–k) suggests a mechanism for switching in the switch$_{ΔPAA}$ mutant (Fig. 4a). Loss of the PAA motif shortens the FliG$_{L3}$ linker and adjusts the position between FliG$_{D3}$ and FliG$_{D4}$. It also moves FliG$_{D2}$ to eliminate a domain swap. Finally, the PAA motif in FliG$_{D3}$ normally interacts with the first helix of FliM$_{mid}$, which is held under tension in the CCW pose[51]. Removing this interaction allows FliM to rotate. To support this proposal, we mapped directionally biasing mutants[52] onto the CCW structure (Fig. 4b). CCW-biasing mutations group to a surface suggested to be a CheY binding site[51,53,54]. These likely disrupt CheY binding to reduce CW transitions. Conversely, CW-biasing mutations dominate the ~100 Å pathway that connects the N-terminus of FliM$_{mid}$ to FliG$_{D5}$ (Fig. 4a,b). These mutations may prevent allosteric conformational changes of FliM and FliG.

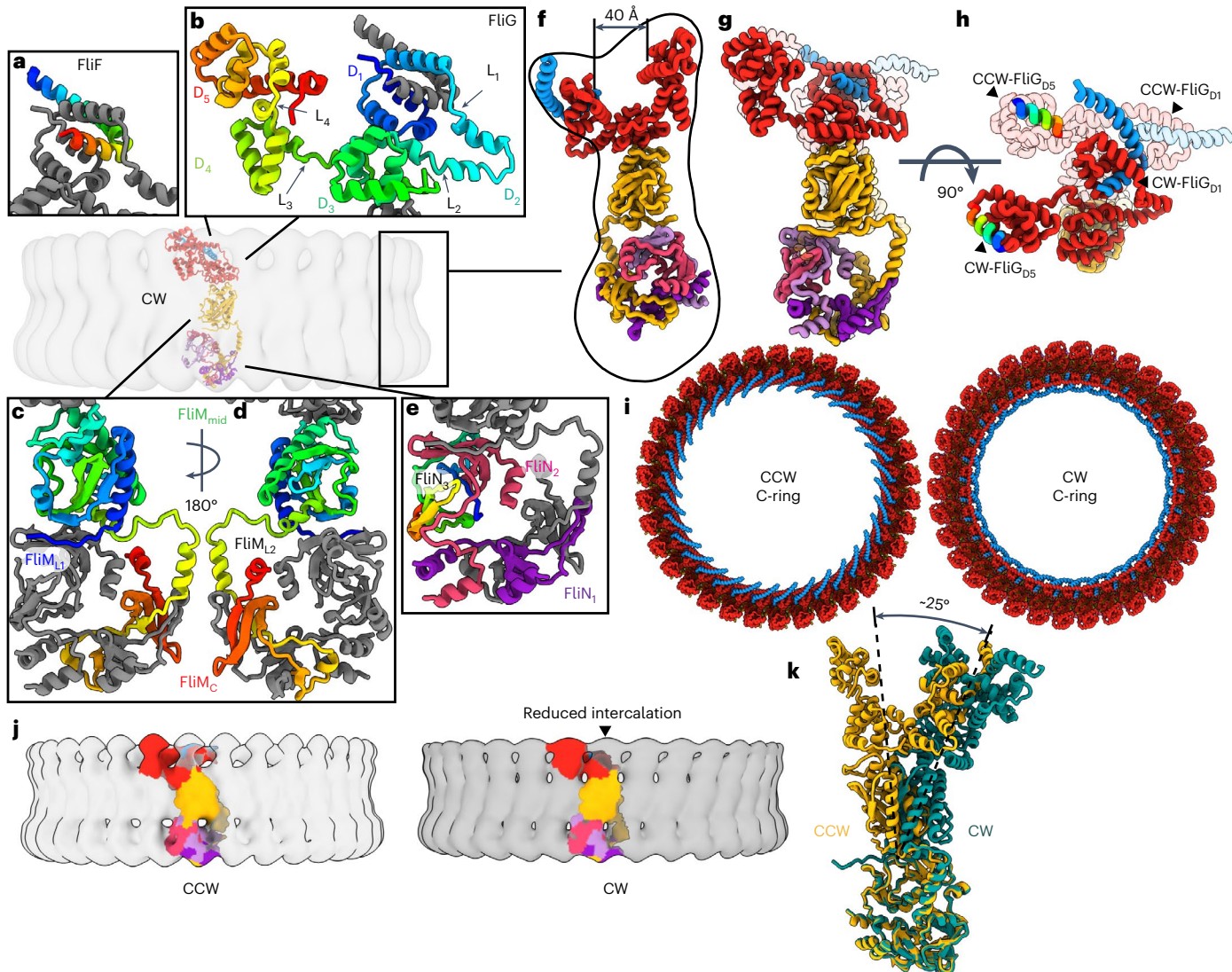

**Fig. 3 | The CW pose of the switch. a–e**, Individual folds of the C-ring subunits in the CW pose: FliF (**a**), FliG (**b**), FliM (**c**), a 180° rotated view of FliN (**d**), FliN (**e**). The relative orientation and the colouring are the same as for Fig. 2. **f**, A side view of the CW pose showing an expanded cleft between $FliG_{D1/D2}$ and $FliG_{D5}$. **g**, Comparison of the CCW pose (transparent) with the CW pose (solid) of a single C-ring subunit. **h**, A 90° rotation of panel (**g**) highlights the magnitude of the conformational change. **i**, Comparison of a top-down view of the CCW and CW poses shows the reversed orientation of the $FliF_C$ helix, which changes the connection to the MS-ring. The increased size of the cleft in the upper ring of FliG is also apparent. **j**, Colouring a single subunit in the context of the C-ring highlights the increased domain swaps in FliG of the CCW pose compared to the CW pose. In the CCW pose, $FliF_C$–$FliG_{D1}$ is in the inner ring above FliM and crosses staves three times. In the CW pose, $FliF_C$–$FliG_{D1}$ is in the inner ring behind $FliG_{D5}$ and crosses staves twice. **k**, A side view of a single unit aligned to the $FliM_C$:3$FliN_C$ spiral highlights the 25° rotation of $FliM_{mid}$ in the CW pose.

## The CW pose bound to a regulator

In one of our CW switch$_{\Delta PAA}$ datasets, we observed density for a binding partner within the cleft in the FliG subunit (Fig. 5a,b and Extended Data Fig. 5b). This identifies one way that a regulatory protein might interact with the CW switch. The bound protein moved both $FliF_C$/$FliG_{D1/D2}$ of the inner ring and $FliG_{D5}$ of the outer ring, as a unit, by ~10 Å toward the centre of the ring. This decreased the diameter from 460 Å in the CW pose (470 Å in the CCW pose) to 440 Å in the CW pose with the binding partner.

The size of the density is consistent with a globular domain of ~120 amino acids. Because it bridges $FliF_C$/$FliG_{D1}$ and $FliG_{D5}$, which differ between CCW and CW, this protein should be specific for the CW pose (Fig. 5a,b). Additional density above the cleft resembles an intertwined helical coiled coil and forms a ring. The position of this ring differs from the position of those that appear on the outer perimeter of FliG during assembly and disassembly[55].

The local resolution for this density was 9 Å, making it difficult to identify the species from the maps. We used manual docking to evaluate several possibilities: YcgR, CheY-FliM$_{1–16}$, quinol:fumarate reductase and FliO (5Y6H ref. 56, 4IGA ref. 57, 1KF6 ref. 58, https://alphafold.ebi.ac.uk/entry/A0A5C2LXN8)[29]. YcgR and CheY could be docked, while quinol:fumarate reductase and FliO fit the density poorly. YcgR[22,23] is unlikely because it physiologically stabilizes the CCW pose. While CheY is relevant to the CW pose[16,17], past work identifies that CheY binds to the switch at the N-terminus of FliM (FliM$_{1–16}$)[53], in the central domain of FliM (FliM$_{51–228}$)[54] with FliM$_{R94}$ suggested as key[51] and at a hydrophobic patch of a FliN homodimer[59] containing FliN$_{VI13}$, FliN$_{VI14}$ and FliN$_{AII5}$. Moreover, past tomography identifies that CheY correlates with the appearance of density on the exterior of the FliM subunit[45,46]. Thus, we cannot identify the bound species.

## Torque transmission

Torque transmits between the MotA/B binding site on the $FliG_{D5}$ torque helix[4,26,60] and FliF. To trace the path of torque transmission, we evaluated the two ends of this pathway. In wild-type C-rings, torque enters the C-ring at the outer $FliG_{D5}$ (Fig. 6b). In the CW pose, $FliG_{D5}$ rotates

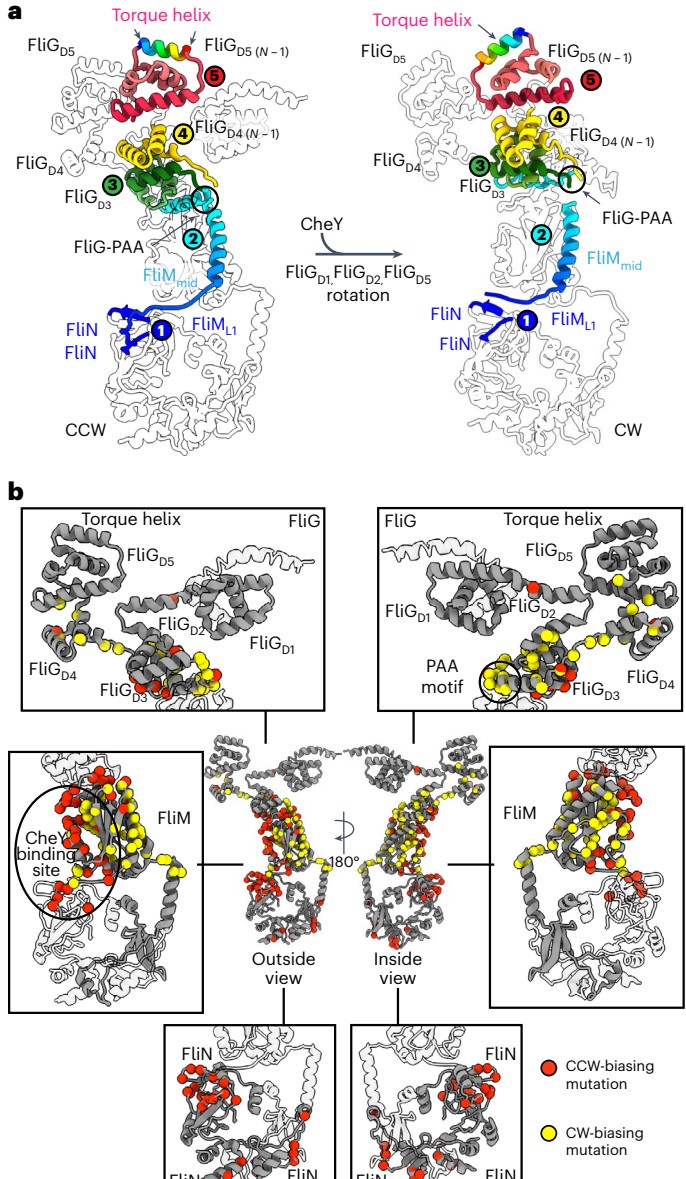

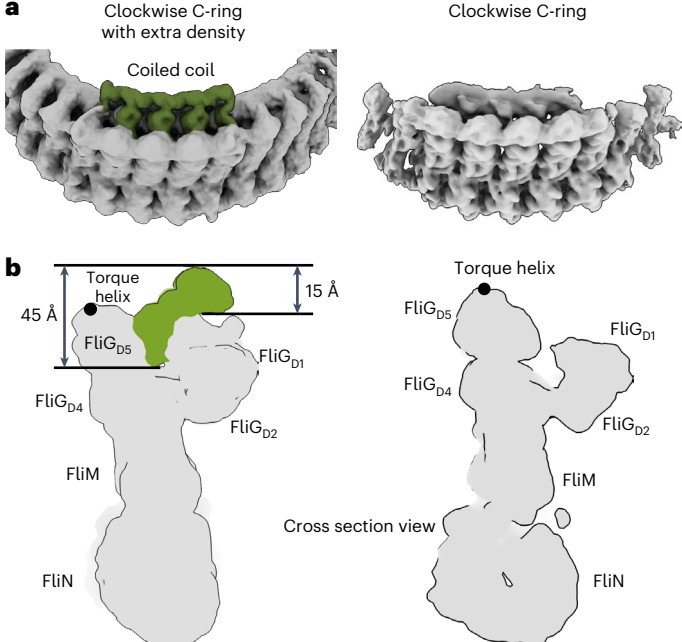

**Fig. 5 | A regulator bound to the CW pose of the switch. a**, On the left, twelve repeats of the protein-bound CW pose of the switch are shown in grey, with three repeats of the density in the cleft shown in green. For comparison, the right shows the CW pose of the switch not bound to a partner protein. The cleft is still visible but lacks density within it. **b**, Cross-section of a single subunit with the density for the regulator in green. For comparison, the CW pose with an empty cleft is shown on the right.

**Fig. 4 | Allostery in the switch. a**, Allosteric pathway from the FliM N-terminus to the torque helix in $FliG_{D5}$. Different steps of signal transmission are coloured from blue to red and numbered. The transfer of information starts at (1) the N-terminus of FliM near the $FliM_{L1}$ linker at the FliM-FliN interface. The information passes through the first helix of (2) $FliM_{mid}$ to $FliG_{D3}$ near the (3) PAA motif, which supports (4) $FliG_{D4}$. A rotation of (5) the C-terminal $FliG_{D5}$ changes the orientation of the torque helix. Note that allosteric signal transmission may involve both the pathway that is shown and a concerted signal transmission in adjacent subunits of the ring. A single subunit of the C-ring with a neighbouring FliG ($FliG_{N-1}$) is shown. **b**, Locations of directionally biased mutations in the switch. Red balls highlight CCW-biasing mutations, with the majority of these located in the proposed binding site for CheY. Their mutation could affect CheY binding. Yellow balls mark locations of CW-biasing mutations, which cluster along the pathway in **a**. Their mutation could release the CCW pose.

and presents the MotA/B binding site to the inside of the ring, with torque likely transmitted along the same path, albeit in the opposite direction. Tracing the path of torque transmission to the rod requires understanding how the switch connects to the MS-ring[1–3]. To evaluate this, we determined the structure of the MS-ring in the CCW switch (Fig. 6a) by obscuring the C-ring using particle subtraction (Extended Data Fig. 1b). The resultant 3.4 Å resolution structure contained 33-mer

MS-rings ($FliF_{50–428}$) in 58% of 34-mer C-rings (Extended Data Fig. 6a,b). The remaining MS-rings could not be classified into a stoichiometry. MS-rings that could not be classified correlated with grids that had thinner ice, suggesting that the MS-ring preferentially partitions at the air–water interface. However, we cannot exclude the presence of other stoichiometries that we could not classify.

While the fold of each FliF subunit is generally consistent with previous reports[5,9–11], there are two notable differences. The first is the 33-mer stoichiometry. Some past structures show variable stoichiometry[5,10]. Others suggest that the native stoichiometry is a 34-mer[9]. A second difference in the MS-ring structure compared to past work was additionally observed regions of N-terminal ring-building motif 1 (RBM1, $FliF_{50–106}$; Extended Data Fig. 6a). Past MS-ring structures identified either 9 or 11 RBM1s[5,9–11], but coordinates were not assigned. The present map contained density for 33 RBM1 domains in two positions and allowed coordinates to be assigned to 11 (Extended Data Fig. 6b).

To complete the connections, we modelled $FliF_{1–49}$ and $FliF_{429–514}$ with AlphaFold[29]. The prediction showed high confidence that these were helices (Fig. 6a). Because the particles have a symmetry mismatch between the MS- and C-rings, there could be multiple ways to model the MSC-ring species. A $FliF_{33}$:$FliG_{34}$:$FliM_{34}$:$FliN_{102}$ assignment is consistent with the symmetric appearance of the C-ring before averaging. By contrast, loss of a FliG subunit, that is, $FliF_{33}$:$FliG_{33}$:$FliM_{34}$:$FliN_{102}$, does not match our data and suggests that the FliF:FliG ratio need not be 1:1. We propose that these MSC-rings contain 203 subunits: 33 FliF, 34 FliG subunits, 34 FliM subunits and 102 non-equivalent FliN subunits.

A global view of the resultant model shows that the MS- and C-rings stack with a tilt angle of 4° (Fig. 6a). This non-coaxial stacking is similar to what was observed between MS-ring and LP-ring in previous structures[6,12] (Fig. 6a). When considering the intact motor (Fig. 6a), it is tempting to align the axis of the C-ring with the LP-ring, which results in a tilted MS-ring between these features. This tilt could explain why the MS-ring looks thicker on edge in low-resolution MSC-ring structures[14,15]

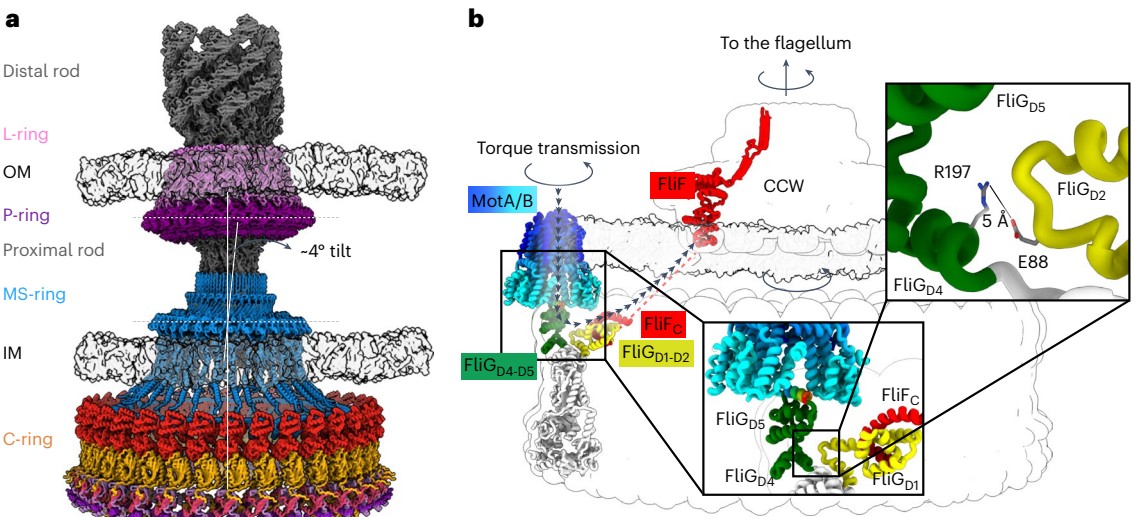

**Fig. 6 | Torque transmission during flagellar rotation. a**, A model for the flagellar motor from Gram-negative bacteria was built from our structure and that of the *S. enterica* flagellar basal body (7CGO ref. 12). The L-ring (light purple) contains FlgH subunits. The P-ring (dark purple) contains the FlgI subunits. In the centre of the LP-ring is a rod (grey). The distal region of the rod (FlgG) connects to the hook and flagellum, while the proximal region of the rod (FliE, FlgB, FlgC and FlgF) connects the LP-ring to the MS-ring. The MS-ring (blue) localizes within the inter-membrane space and contains FliF subunits. Finally, the C-ring (orange), contains FliF C-termini as well as FliG, FliM, and FliN of the switch.

**b**, Pathway of torque transmission through the flagellar motor. The C-ring transmits torque from the MotA/B stator to the MS-ring and flagellar rod. The figure shows a map of the torque transmission pathway highlighted with black arrowheads and coloured from blue (stator) to red (MS-ring). Torque transfer begins with the interaction between the MotA/B stator and the torque helix of FliG$_{D5}$ of the C-ring. Interactions across the FliG subunit allow the torque to be transmitted to FliG$_{D1}$, where there is a direct interaction with FliF. This is expected to turn the MS-ring and the flagellar rod within.

than in structures of the isolated MS-ring[5,9–11]. A tilt could be expected in any motor with a symmetry mismatch between the MS- and C-rings.

With this tilt, axial rotation of the C-ring and rod during flagellar rotation would make the MS-ring of the motor appear to wobble when viewed edge on (Supplementary Video 3). Symmetry mismatches have other characterized biological effects. They promote a low-energy state and allow efficient rotation[61] at high speed, which requires that there be no significant energy minima. Symmetry mismatch between the 33-mer MS-ring and 34-mer C-ring observed in these particles (Fig. 6a and Extended Data Fig. 1) could help support this. This wobble could prevent an energetic minimum during flagellar rotation[10] or influence the inherently asymmetric export apparatus[62,63] at the interior of the C-ring. Because FliF$_C$ is tightly tethered to FliG$_{D1}$, the ability of the MS- and C-rings to be flexibly attached may be important for both torque transmission and the shift between the CCW to CW poses.

## Discussion

In this Article, we report cryoEM structures of three states of the flagellar switch, answering questions about torque transmission, directional control and binding of a response regulator. Torque input (Fig. 6b) involves the MotA/B stator[1–3] (Extended Data Fig. 7a,b), which uses the transmembrane electrochemical gradient to induce rotation and transmits this to the torque helix on FliG$_{D5}$ (refs. 4,60,64–66). MotA/B and the C-ring then act like interlaced cogwheels to drive flagellar rotation[64,65] (Extended Data Fig. 7c). Supporting this model, the interface between MotA subunits[64,65] is perfectly positioned to grasp the FliG$_{D5}$ torque helix (Extended Data Fig. 7a,b). Torque transmits through FliG to FliF of the MS-ring, which connects to the flagellar rod.

FliG has a conformational difference between CCW and CW poses (Figs. 2 and 3) that moves the binding site for MotA/B from facing outward in the CCW pose to facing inward in the CW pose (Fig. 3), which changes the direction of rotation of the C-ring (Supplementary Video 4 and Extended Data Fig. 7c). This model is supported by tomography of the stator bound to the C-ring in the CCW and CW poses[46] and is consistent with past proposals for powering rotation in opposite directions[64,65]. The concomitant ~180° FliF$_C$/FliG$_{D1}$ rotation optimizes the

connection between the C-ring and the MS-ring in these two directions (Fig. 3i).

The CCW and CW poses of the switch also help explain cooperativity[49,50], which induces the subunits of the C-ring to preferentially adopt the same pose. Three conformational differences between the CCW and CW would result in steric clash unless the next subunit adjusted. These are the FliF$_C$–FliG$_{D1}$ rotation, the change in domain swapping of FliG$_{D2}$ and the ~25° rotation of FliM$_{mid}$ (Fig. 3g,h,k). Biologically, bound signalling proteins influence both the direction and rate of rotation. One of our CW data sets showed density within an ~40 Å cleft between FliG domains, identifying one binding site for regulatory proteins (Fig. 5). This density is consistent with an ~120 amino acid protein that locks FliF$_C$–FliG$_{D1/D2}$ and FliG$_{D5}$ into the CW pose. The ~30 Å cleft in the CCW pose may also be large enough to bind to a regulatory protein (Fig. 2f).

The architectures of FliM and FliN also inform on the mechanism of conformational transitions between the CCW and CW poses[45,46] (Figs. 2–4). It has previously been unclear how the motor could be conformationally dynamic enough to function and yet stable enough to survive these large structural transitions. Domain swapping with subunit intercalation is a structural feature known to both enhance stability and allow superstructures to dynamically adopt multiple stoichiometries. This has been best studied during protein aggregation that causes disease[67]. The substantial domain exchange of FliM and FliN within the switch may help to accommodate large molecular reorganization during the transitions between the CCW and CW poses (Figs. 2–4).

Notably, there is currently no consensus over the stoichiometry for the MS-ring[5,9,10,12]. Some cryoEM studies showed a range of stoichiometries[5,10]. Other studies only identify a 34-mer[9]. The variable stoichiometry was interpreted as the MS-ring adapting to load. This largely leveraged parallels to the C-ring's stoichiometry[68] and the number of bound stators[69], which can change in response to the strength of the attractant or load.

Studies showing only a 34-mer suggest that other stoichiometries arise from artefacts due to C-terminal proteolysis of FliF or incorrect templating during plasmid expression[9]. We can exclude C-terminal

proteolysis affecting stoichiometry in our structure because we observe density for the full C-terminus of FliF bound to FliG (Figs. 2a and 3a and Extended Data Fig. 3a). In terms of templating, this 33-mer MS-ring and the previously published strict 34-mer[9] were similarly expressed in *E. coli*. This suggests that the *E. coli* templating machinery can be recruited to assemble the *Salmonella* MS-ring (~95% identical) and is also unlikely to underlie the stoichiometric difference. Nevertheless, we did not test conditions proposed to affect stoichiometry, which would be required to distinguish between an adaptive and a strict stoichiometry. For example, we did not coexpress the *S. enterica* templating machinery with the pKLR3 plasmid, and we did not grow cells under conditions with different attractants or different loads. Taken together, the origins of symmetry differences in MS-ring structures remain unclear at this time.

In aggregate, this work reports the high-resolution structure of the most critical piece of the flagellar motor in three states. The structure suggests mechanisms for torque transmission and directional switching during chemotaxis in *Salmonella* and related bacteria. This structure also allows a large body of data on bacterial chemotaxis to be understood in the context of an architecture.

## Methods

### Constructs

Plasmid pKLR3 (ref. 27) containing the *S. enterica* serovar *typhimurium fliL*, *fliF*, *fliG*, *fliM*, *fliN* and *fliO* genes was a generous gift to M.E. from S. Khan.

### Protein purification

The *S. enterica* serovar *typhimurium* FliFGMN subunits were expressed in *E. coli* BL21-Gold cells in LB medium supplemented with 0.034 mg ml$^{-1}$ chloramphenicol. At an $OD_{600}$ of 0.6, expression was induced with 1 mM isopropyl β-D-1-thiogalactopyranoside. Following induction, cells were grown at 37 °C for 18 h with shaking, then collected by centrifugation at 6,750 × *g* at 4 °C.

Following growth, bacterial cells were resuspended in 100 mM Tris–HCl with pH 8.0, 8 mM EDTA and one protease inhibitor tablet for every 50 ml of buffer. Cells were lysed with 1% *w/v* Triton X100 detergent and 10 mg lysozyme. The cell suspension was stirred at 4 °C for 4 h before adding $MgCl_2$ to a final concentration of 10 mM. For every 50 ml of buffer, 100 U of DNase and 5 mg of RNase were also added. The lysed cells were stirred for 1 h before centrifuging at 18,000 × *g* for 30 min to remove cell debris. Membranes were separated from this supernatant by centrifugation at 60,000 × *g* for 1 h. The cell membranes were resuspended in 100 mM HEPES with pH 7.5, 5 mM EDTA, 0.1% *v/v* Triton X100 for 30 min on ice. Partial purification was achieved via differential membrane extraction. First, the Triton X100 concentration was increased to 10%, and the suspension was mixed gently for 1 h. These partially extracted membranes were centrifuged at 14,000 × *g* for 30 min, and the supernatant was centrifuged at 60,000 × *g* for 1 h to collect the membranes. This pellet was resuspended in 100 mM HEPES with pH 7.5, 5 mM EDTA, 0.1% *v/v* Triton X100 and 0.05% lauryl maltose neopentyl glycol (Anatrace, NG310), then mixed gently for 1 h. The suspension was then spun at 14,000 × *g* for 30 min, and the supernatant was filtered through a 0.4 μm syringe filter. The filtered sample contained MSC-ring particles and was used for preparing cryoEM grids.

### Cryo-EM sample preparation and imaging

A 300 mesh R1.2/1.3 Au Quantifoil grid (Electron Microscopy Sciences) was glow discharged for 15 s. Purified MSC-rings (2 μl of 23 mg ml$^{-1}$) were added to each grid at 4 °C and 100% humidity. After 15 s of incubation, blotting was performed for 4 s. Grids were plunged into liquid ethane using a Vitrobot Mark IV system (Thermo Fisher). Grids were screened on 200 keV Glacios microscope (Thermo Fisher). Data were collected from the best grids using a 300 keV Titan Krios G4 microscope with a Gatan K3 direct electron detector (Thermo Fisher).

For wild-type MSC-ring, 34,831 movies were motion-corrected using patch motion-based correction in cryoSPARC (v.4.2.1)[70]. The contrast transfer function (CTF) was estimated using Patch CTF Estimation in cryoSPARC[70]. Using template picker, we picked 3,906 particles from 3,427 micrographs (10% of the dataset). An ab initio model was created from this and was used as a template to pick particles from the complete dataset.

For the CW MSC-ring, 35,552 movies were first motion-corrected using patch motion-based correction in cryoSPARC[70]. The CTF was estimated using Patch CTF Estimation in cryoSPARC. Using template picker, we picked 3,906 particles from 3,500 micrographs (10% of the dataset). An ab initio model was created from this and was used as a template to pick particles from the complete dataset.

For the CW MSC-ring with bound partner protein, 26,130 movies were processed using the same steps as for CW C-ring. Using cryoSPARC template picker, we picked 4,546 particles from 3,700 micrographs (14% of the dataset). Using the CW C-ring as an input model, the C-ring was created and was used as a template to pick particles from the complete dataset.

### Structure determination

For the wild-type MSC-rings, 295,031 particles were picked, and 50,000 of them were used to build ab initio models without enforcing symmetry. These initial 3D class averages separated into four classes. In the next step, all 295,031 particles were used to perform heterogeneous refinement on the four ab initio classes. Upon completion, only one class contained MSC-rings (21% of particles; Extended Data Fig. 1a). In addition, one class was isolated MS-rings, and two classes were junk classes. Inspection of the density showed the C-ring at low resolution, ~15 Å, and showed clear staves. However, the associated density for the MS-ring was uninterpretable.

Because the lower quality density for the MS-ring may have resulted from a symmetry mismatch between the MS- and C-rings, potentially combined with alignment that was biased toward the larger C-ring, a particle subtraction technique was used to improve the resolution. First, MS-rings were identified from the 3D model. This consensus map was used to build a mask around the MS-ring. The MS-ring was then subtracted at the level of the micrograph (Extended Data Fig. 1a). The MS-ring-subtracted particles were then used to determine high-resolution structures for the C-ring.

Initial heterogeneous refinement resulted in five classes with different symmetries. One class contained 33-fold symmetry (13,081 particles), two classes contained 34-fold symmetry (15,301 and 13,081 particles), one class contained 35-fold symmetry (11,346 particles) and one class contained 36-fold symmetry (6,041 particles). Non-uniform refinement[3] was performed on each of the above classes (33-, 34-, 35- and 36-fold symmetry) with the final resolution of the C33 map at 4.5 Å resolution, the C34 map at 4.1 Å resolution, the C35 map at 4.5 Å resolution and the C36 map at 6.7 Å resolution. To improve the resolution of the 34-fold symmetric map even further, particles from all the classes were passed through heterogeneous refinement with C34 symmetry imposed. Following this procedure, the best classes were refined using homogeneous and non-uniform refinement. The final resolution was 4.0 Å.

For the CW MSC-rings, 43,741 particles were picked. Using the ab initio model, these particles were classified into three classes. Upon completion, only one class contained MSC-rings (16% of particles; Extended Data Fig. 5a). In addition, one class was isolated MS-rings, and one class was junk class. Inspection of the density showed the C-ring at low resolution, ~20 Å, and showed clear staves. However, the associated density for the MS-ring was uninterpretable. From the CCW C-ring previous knowledge, we applied C34 symmetry to refine the C-ring. Due to the low number of particles, we were unable to improve the resolution of the map using particle subtraction. Therefore, we expanded the particles by applying C34 symmetry and locally refined

a small section by masking three staves of the C-ring. This improved the resolution to 4.6 Å. We used RELION (v.4.0.1) image handler to form a 34-mer C-ring from the refined sectional map[71].

For the CW MSC-rings with bound partner protein, 59,404 particles were picked. Using the CW MSC-ring model, these particles were classified into three classes. Upon completion, only one class contained MSC-rings (34% of particles; Extended Data Fig. 5b). In addition, one class was isolated MS-rings, and one class was a junk class. Inspection of the density showed the C-ring at low resolution, ~25 Å, and showed clear staves. However, the associated density for the MS-ring was uninterpretable. We then applied C34 symmetry to refine the C-ring. As we had more particles in this dataset, we both applied particle subtraction to improve the map quality and also applied local refinement on C34 symmetry-expanded particles from three masked staves. This improved the resolution to 5.9 Å. We used RELION image handler to form a 34-mer C-ring from the refined sectional map[71].

To evaluate the MS-ring associated with CCW C-rings, the particle subtraction protocol was reversed. The 34-mer class has the most particles and the highest resolution. Thus, 34-mer particles were next extracted from the micrographs, and the C-ring was then subtracted (Extended Data Fig. 1b). Classes were then developed for the MS-ring. This procedure identified that for the 34-mer C-ring, the MS-ring classified exclusively as a 33-mer (Extended Data Fig. 1b) and was associated with a final resolution of 3.4 Å.

### Model building and refinement

To assist in model assignment, alphaFold[29] was used to develop homology models of appropriate subunits and domains. For FliG domains, FliG$_{D3}$:FliM$_{mid}$ heterodimers, FliN$_C$ homodimers and FliM$_C$:FliN$_C$ heterodimers, packing interactions from crystal structures of various unassembled domains from thermophiles were used to create a library of multi-domain models with potential domain exchanges. This library was manually developed using COOT (v.0.9.8.8)[72].

The model of FliG$_{D3}$:FliM$_{mid}$ (developed from PDB entry 4FQ0 (ref. 32))was first docked into the corresponding density in COOT, then optimized in ChimeraX (v.1.7)[73]. This was followed by docking the models for FliG$_{D4}$ and FliG$_{D5}$ into the density at the exterior of the ring (developed from PDB entries 3HJL (ref. 38) and 1LKV (ref. 30)). Next, a model containing the FliF$_C$, FliG$_N$ and the domain-swapped armadillo repeat (developed from PDB entry 5WUJ (ref. 36)) was docked into the density on the interior of the ring in COOT[72] and optimized in ChimeraX[73]. Linkers between these domains were built manually in COOT[72]. The base of the ring used a combination of docked FliN$_C$ homodimers and FliM$_C$:FliN$_C$ heterodimers (developed from PDB entry 4YXB (ref. 39)), with the linking helix between FliM$_{mid}$ and FliM$_C$ built manually in COOT[72].

Docking of each of these structures in COOT[72] followed by optimization in ChimeraX[73] gave an unambiguous match with the density. Refinement was performed by standard methods that alternated rounds of manual model improvement in COOT[72] with refinement in PHENIX (v.1.20.1-4487)[74]. Figures were made using ChimeraX[72], and videos were made using Blender v.3.5 (https://www.blender.org/) and Molecular Nodes (v.2.8)[75].

### Manuscript editing using artificial intelligence

Manuscript length and accessibility were both edited using the formalizer subroutine in goblin.tools.

### Reporting summary

Further information on research design is available in the Nature Portfolio Reporting Summary linked to this article.

### Data availability

All raw, processed, and interpreted data that support the findings of this study are available in public repositories. Raw micrographs have been deposited with EMPIAR[76] (https://www.ebi.ac.uk/empiar/), and accession codes are EMPIAR-11597, EMPIAR-11891 and EMPIAR-11892. CryoEM maps have been deposited at the EMDB[77] (https://www.ebi.ac.uk/emdb/) with the accession codes EMD-41100, EMD-41101, EMD-41102, EMD-41103, EMD-41104, EMD-43256, EMD-43258, EMD-43327 and EMD-43328. Atomic coordinates of the 34-mer CCW C-ring and the 33-mer MS-ring have been deposited at the Protein Data Bank[78] (www.rcsb.org) with the accession codes 8T8O and 8T8P. Atomic coordinates for the single subunit of the isolated CW-locked C-ring are deposited with the accession code 8VIB, the 34-mer isolated CW-locked C-ring are deposited with accession code 8VKQ. Coordinates for a single subunit of the CW-locked C-ring bound to a partner protein have the accession code 8VID, and the 34-mer of the CW-locked C-ring bound to a partner protein have the accession code 8VKR. Previously reported structures or computational models used to support this work are: *Thermotoga maritima* FliG (1LKV ref. 30; 5TDY ref. 34), *Helicobacter pylori* FliG (3USW ref. 31, 4FQ0 ref. 32), *T. maritima* FliN (1YAB ref. 42), *S. enterica* FliM:FliN fusion (4YXB ref. 39), *S. enterica* flagellar basal body (7CGO ref. 12), *Aquifex aeolicus* MotA (8GQY ref. 79), *Campylobacter jejunji* MotA/B (6YKM ref. 65), *Clostridium sporogenes* MotA/B (6YSF ref. 64), *Bacillus subtilis* MotA/B (6YSL ref. 64), *E. coli* YcgR (5Y6H ref. 56), *T. maritima* CheY-FliM$_{1-16}$ (4IGA ref. 57), *E. coli* quinol:fumarate reductase (1KF6 ref. 58) and FliO (alphafold.ebi.ac.uk/entry/A0A5C2LXN8[29]). Source data are provided with this paper.

### Code availability

No custom code was used or developed for the analysis of data reported in this study.

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

## Acknowledgements

We thank S. Khan, currently at the Molecular Biology Consortium, Lawrence Berkley National Laboratories and Lahore University of Management Sciences, for the pKLR3 plasmid; T. Nakagawa for technical advice during the early stages of this work; and B. Butler for experimental assistance. We thank W. Chiu and the S2C2 workshop for training in cryoEM. We thank B. DeBuyser for the artificial intelligence-based goblin.tools. This project was funded by National Institutes of Health (NIH) grant GM61606 awarded to T.M.I. and G.C. G.C. is the recipient of a Senior Research Career Scientist award 1K6BX004215 from the Department of Veterans Affairs. M.H.G. is supported by NIH T32 GM007628. Negative stain and cryoEM data were collected at the Center for Structural Biology CryoEM facility at Vanderbilt University. The Glacios cryo-TEM used for screening was acquired by NIH grant S10 OD030292-01.

## Author contributions

P.K.S. expressed and purified protein, determined the structures, analysed the data and wrote the manuscript. P.S. and M.H.G. assisted in the CW structure and data analysis. E.M. and O.A. analysed data. M.E. and G.C. conceived the research, provided scientific advice and contributed to manuscript preparation. T.M.I. conceived the research, supervised the project, assisted in data interpretation, assisted in modelling, performed data analysis and wrote the manuscript with PKS.

## Competing interests

The authors declare no competing interests.

## Additional information

**Extended data** is available for this paper at https://doi.org/10.1038/s41564-024-01674-1.

**Correspondence and requests for materials** should be addressed to T. M. Iverson.

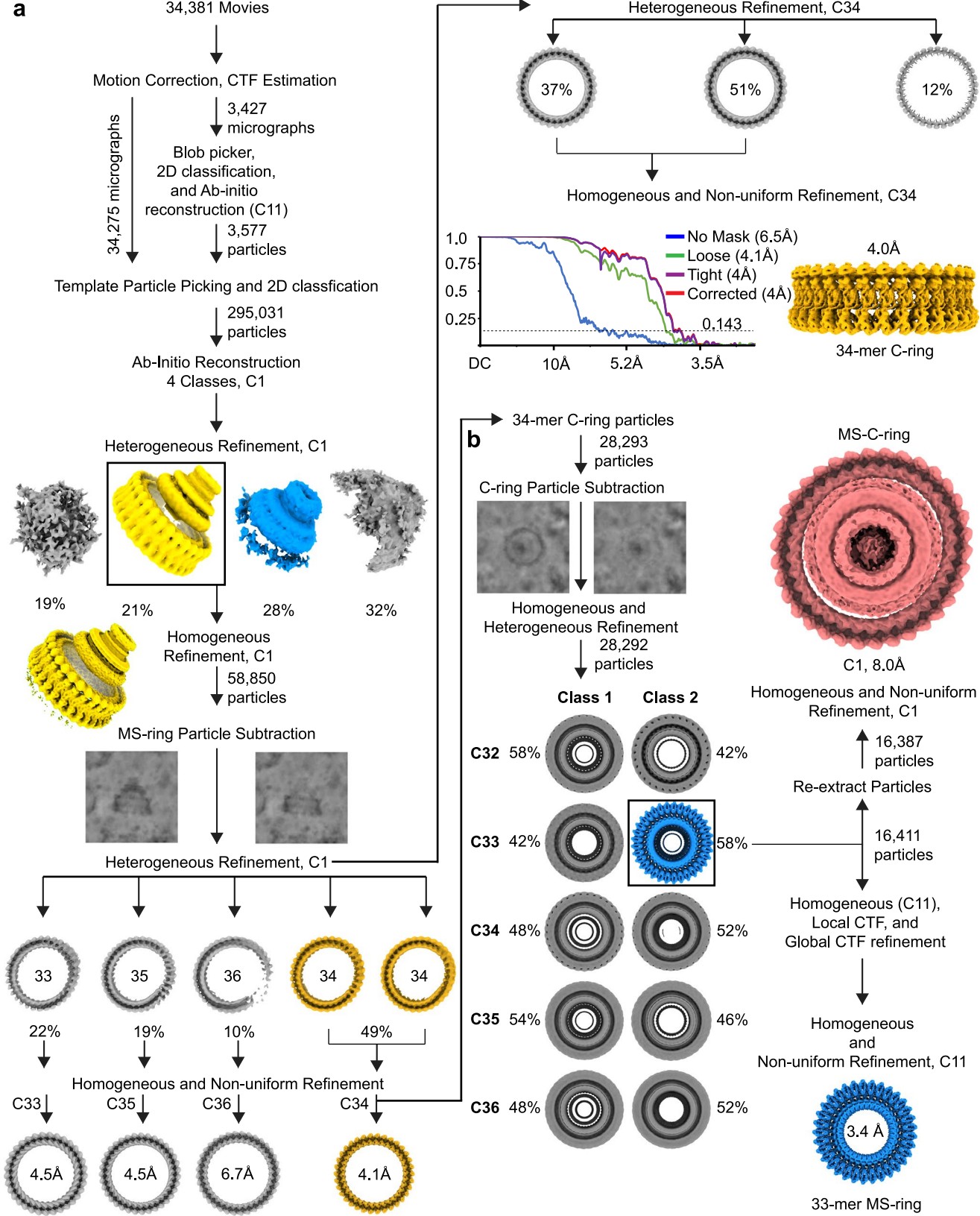

**Extended Data Fig. 1 | See next page for caption.**

**Extended Data Fig. 1 | CryoEM workflow for the CCW C-ring. (a)** Workflow for the reconstruction of the 34-fold symmetric counterclockwise C-ring at 4.0 Å resolution. Following 2D classification, only particles containing both MS- and C-rings were retained. Symmetry could not be unambiguously classified from the combined MS- and C-rings. To identify the structure of the C-ring, MS-rings were removed through particle subtraction from the micrograph. This was saved as a separate dataset so that the original micrographs were retained. Classification of the separate C-ring revealed rings with 33- to 36-fold symmetry dominated by a prevalent of a 34-mer (~50% of particles). The 34-mer C-ring was subjected to C34 heterogeneous refinement and had a final overall resolution of 4.0 Å after this procedure. **(b)** To refine the associated MS-ring, the particles containing 34-mer C-rings were re-identified in the original micrographs and the C-ring was subtracted from these 34-mer particles. The subsequent heterogeneous refinement process used five parallel calculations to individually impose multiple C33-C36 symmetries. Among these, only the C33 MS-ring refinement

(58% of particles) yielded distinct secondary structure features. The remaining particles (42%) did not classify as any observable symmetry. To ensure that we had not missed a subset of MS-rings with other stoichiometries, we removed the C33 particles from the calculation and separately imposed C34, C35, and C36 symmetry. This did not result in a class with interpretable density. To identify why 42% of the MS-rings that were bound to C-rings could not be classified, we re-evaluated the raw micrographs. We identified that the MS-rings that could not be classified were associated with micrographs that had thinner ice, suggesting a preferential orientation with the MS-ring at the air-water interface. We cannot, however, exclude that other symmetries exist at lower abundance in our samples or in the biological system. Notably, C33 symmetrization revealed the RBM3 and β-collar, but masked the details of RBM1 and RBM2. To achieve high-resolution insights into all domains, we subsequently conducted refinement using C11 symmetrization.

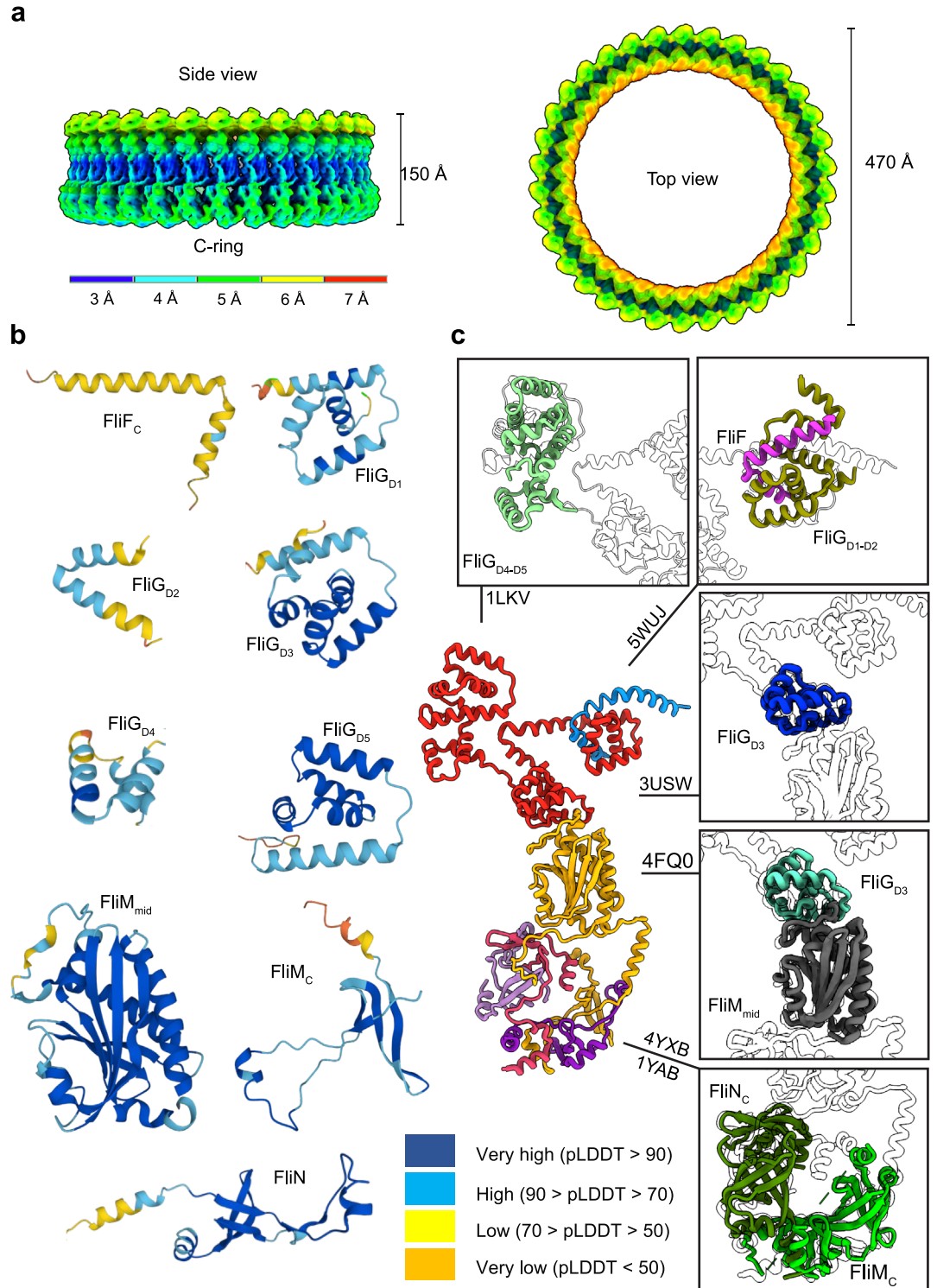

**Extended Data Fig. 2 | Assignment of the CCW pose of the C-ring.** (**a**) Surface representation of the C-ring colored by local resolution. The highest resolution (2.9 Å) is in blue, and the lowest resolution (6.6 Å) is in red. (**b**) AlphaFold models of individual domains were used as starting points in building the structure. The structures are colored by confidence from blue (very high confidence) to orange (very low confidence). (**c**) Comparison of the final model from the cryoEM structure to isolated domains from crystal structures of homologs. Insets show different regions of the structure. FliG$_{D4-D5}$ is superposed with the equivalent

domains from *Thermotoga maritima* (rcsb.org/structure/1lkv[30]). FliG$_{D3}$ is shown superposed with the equivalent domain from *Helicobacter pylori* (rcsb.org/structure/3usw[31]). FliG$_{D1-D2}$ and FliF$_C$ are shown superposed with the equivalent domains from *T. maritima* (rcsb.org/structure/5tdy[34]). The FliM$_{mid}$ domain and FliG$_{D3}$ are superposed with the equivalent domains from *H. pylori* (rcsb.org/structure/4fq0[32]). FliM$_C$ and FliN are superposed with the FliN dimer from *T. maritima* (rcsb.org/structure/1yab[42]) and the fused FliM$_C$-FliN$_C$ dimer from *Salmonella enterica* (rcsb.org/structure/4yxb[39]).

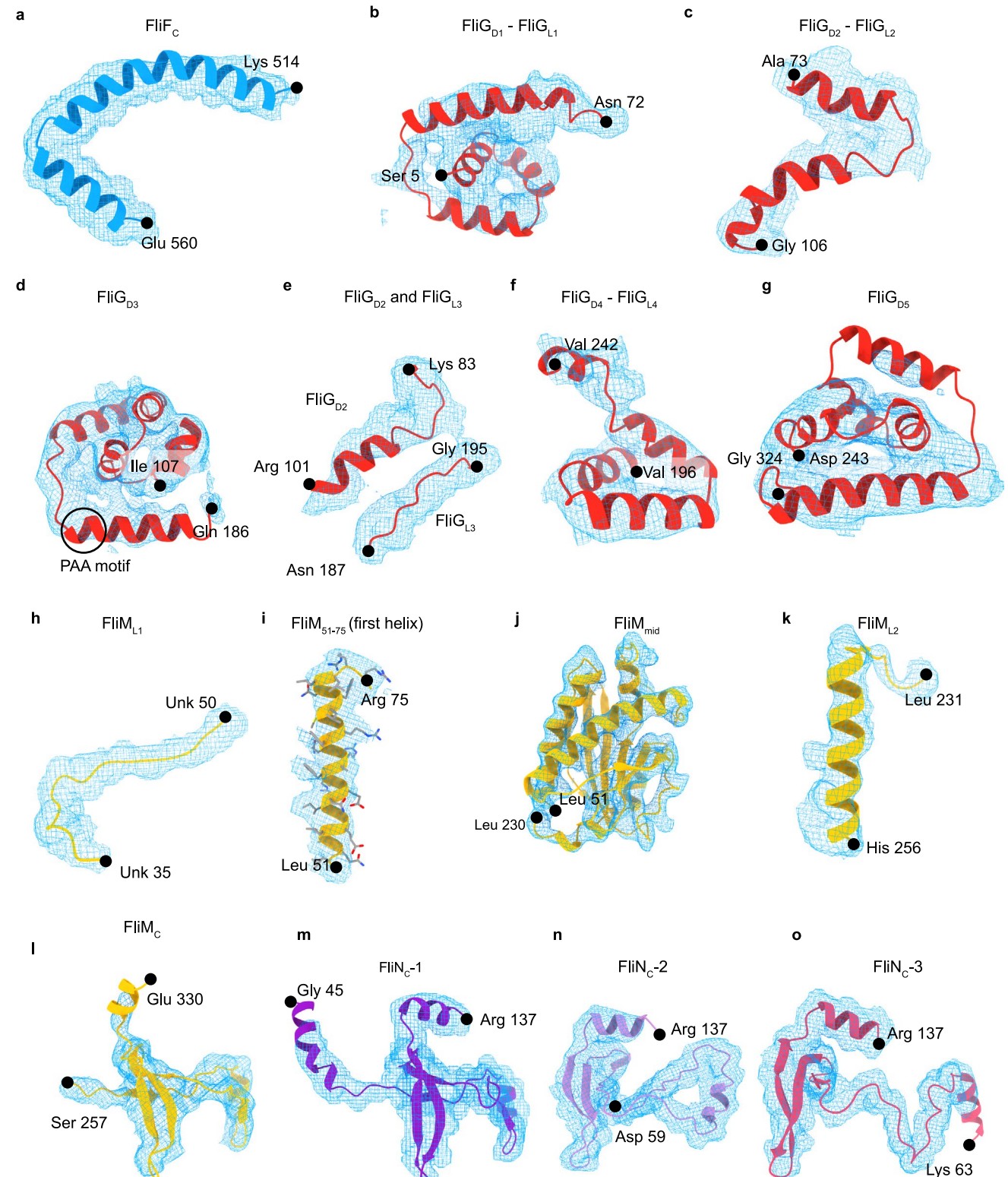

**Extended Data Fig. 3 | Density for subunits of the C-ring in the CCW pose.** Density for: (**a**) FliF$_C$; (**b**)−(**g**) regions of the FliG subunit; (**h**)−(**l**) regions of the FliM subunit (**m**)−(**o**) each of the three FliN$_C$ domains. The resolution of FliM$_{mid}$ is < 3 Å. In panel (**i**), the side chains are shown for FliM$_{51-75}$ to highlight the map quality.

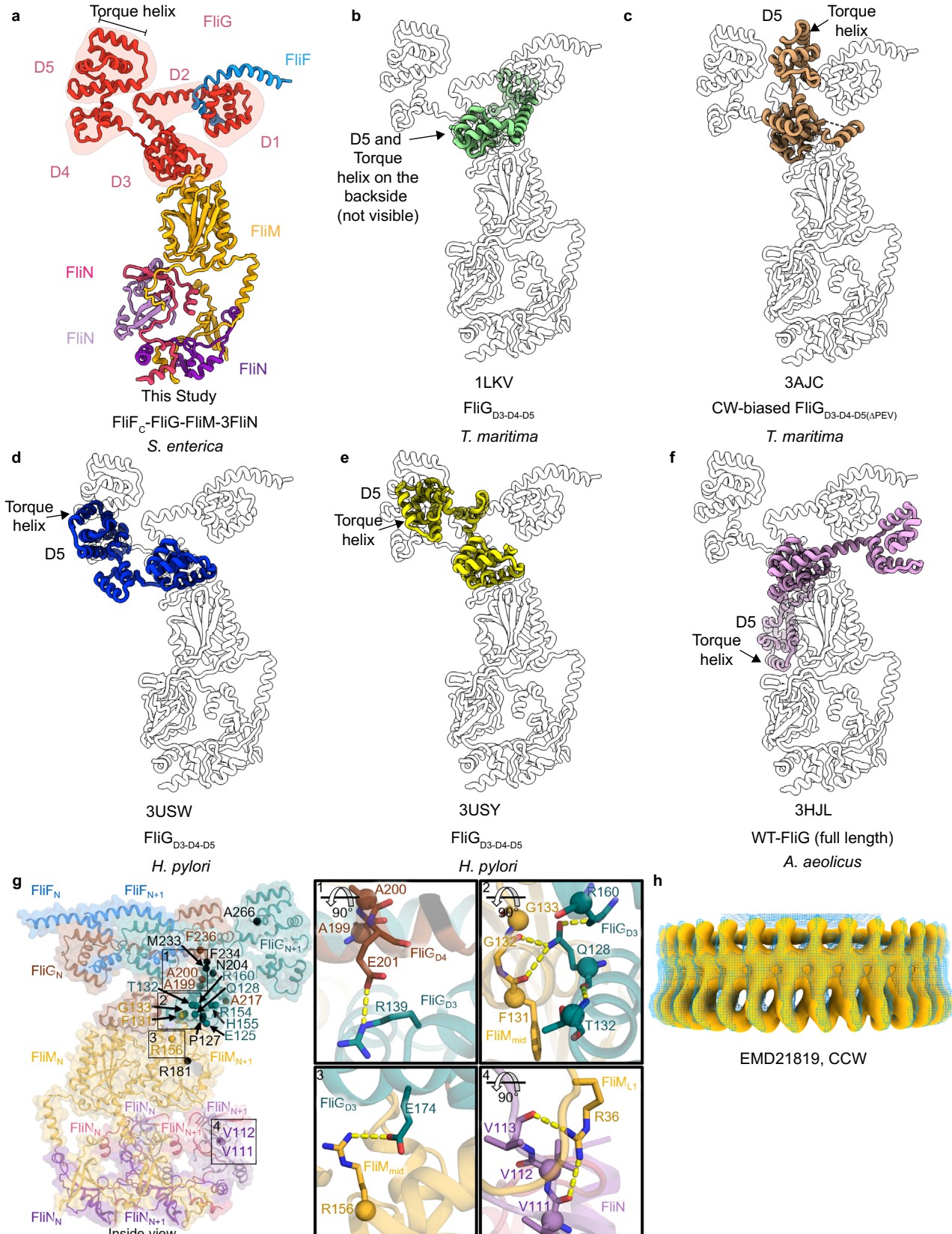

**Extended Data Fig. 4 | Validation of the assembled C-ring.** (**a**)–(**f**) Individual domains of the FliG subunit concur with the domains of this structure. However, the global appearance of FliG differs due to the different interdomain angles. (**g**) Locations of flagellum-deficient mutations[30,31,40,80] in the context of two adjacent protomers of the C-ring. All flagellum-deficient mutations are highlighted with a sphere. Flagellum deficient mutations that map to the interior domains and are likely to prevent flagellar assembly through misfolding of an individual subunit is colored black. Flagellum deficient mutants that interact with adjacent protomers in the ring are the same color as the associated chain. Insets highlight key select intersubunit interactions (yellow dash) that may be disrupted with these mutations. (**h**) comparison between the CCW pose and a CCW tomogram from *Vibrio algintolyticus*[45].

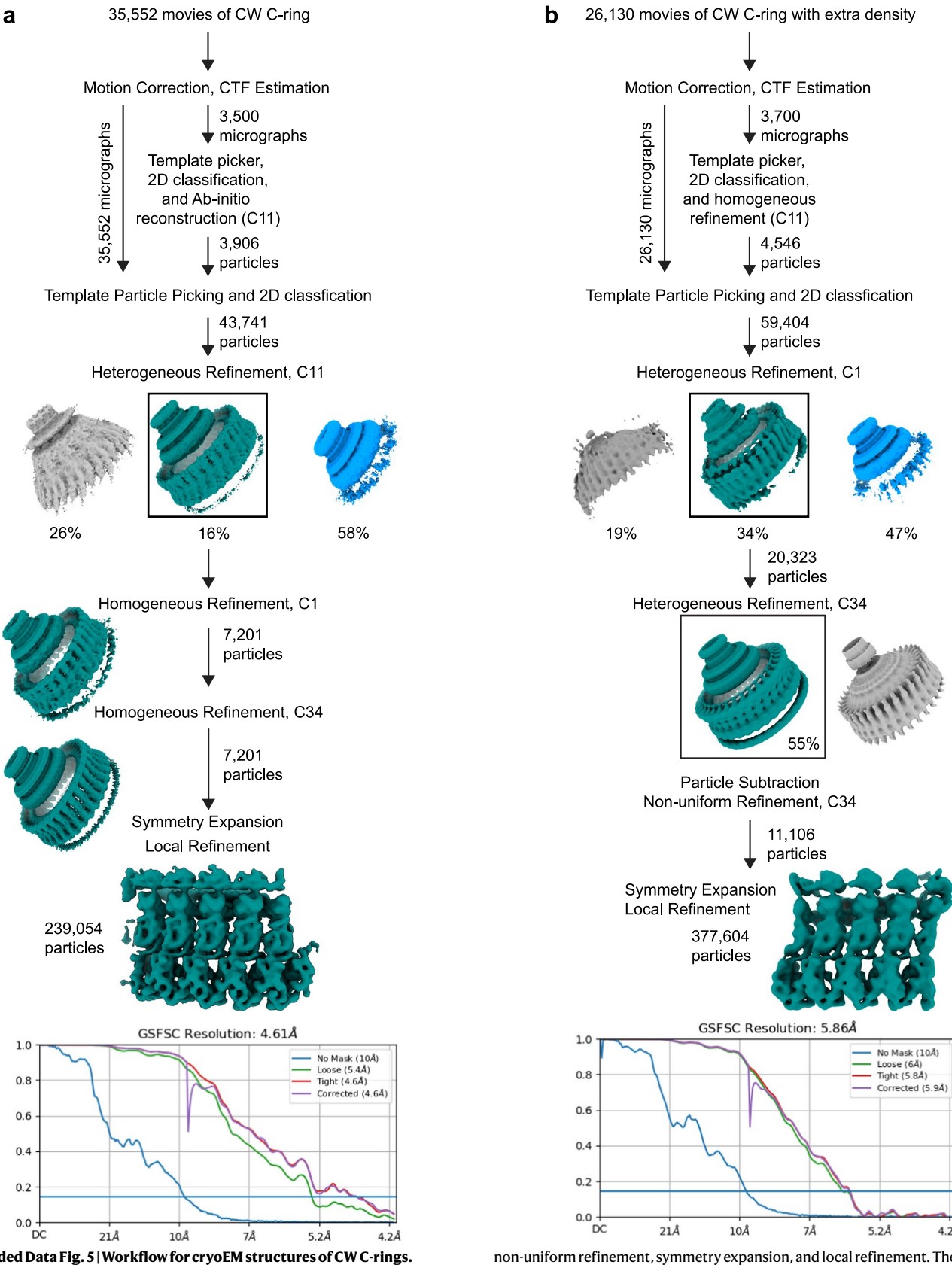

**Extended Data Fig. 5 | Workflow for cryoEM structures of CW C-rings.**
(**a**) Workflow for the unbound CW C-ring at 4.6 Å resolution. Representative
3D reconstructions used symmetry expansion followed by local refinement.
(**b**) Workflow for the CW C-ring with a bound partner at 5.9 Å resolution.
Representative 3D reconstructions used particle subtraction followed by
non-uniform refinement, symmetry expansion, and local refinement. The raw
micrographs for CW rings all had thinner ice than the micrographs for CCW rings.
Because of this, the MS-ring structure could not be classified in any case, and the
structure could be determined without the application of particle subtraction.

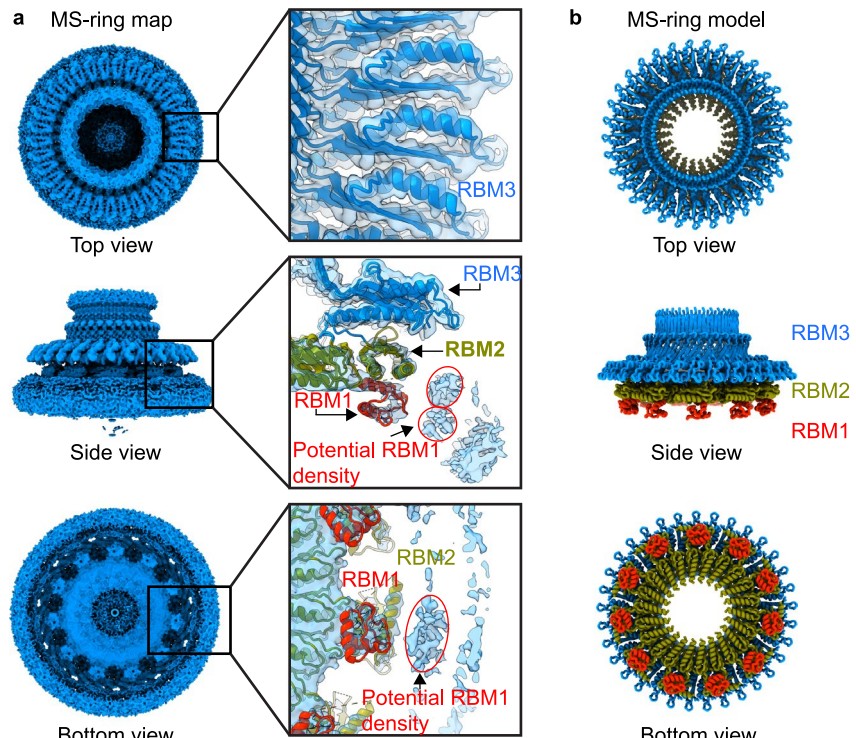

**Extended Data Fig. 6 | Structure of the MS-ring.** (**a**) The MS-ring is composed entirely of copies of the FliF subunit. Past structures have been determined with different stoichiometries[5,6,9,10,12], although there remains debate on whether there is a biologically-relevant exact stoichiometry. Three views of representative cryoEM density (*blue* mesh) for the 33-mer MS-ring, calculated at 3.4 Å resolution and superposed onto the final model. RBM1 is *red*, RBM2 is *olive* and RBM3 is *blue*. Regions of density corresponding to other positions of RBM1 but where the quality was not sufficient to assign are circled. (**b**) Model of the MS-ring highlighting the relative positions of the ring-building motifs.

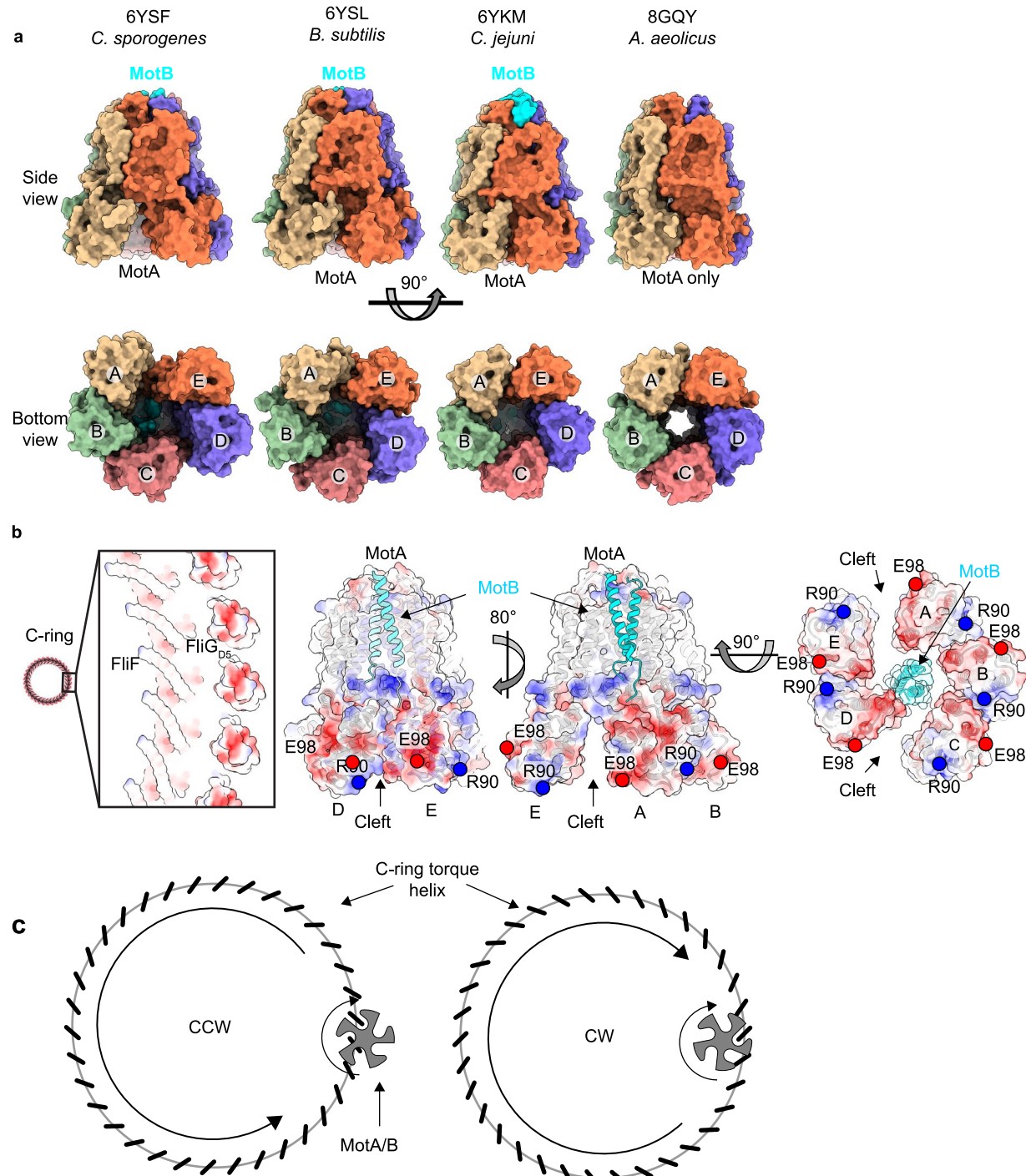

**Extended Data Fig. 7 | Bidirectional rotation of the C-ring by MotA/B.**
MotA subunits are labeled A – E in the panels. (**a**) Comparison of MotA/B[64,65] and MotA[79] cryoEM structures identifies varying levels of asymmetry that change the width of the cleft between subunits. The most symmetric structure is that of isolated MotA[79]. In the context of past biochemistry and structures of MotA/B, the MotA/B stator binds to the torque helix on FliG$_{D5}$. A compelling model would use a cleft between MotA subunits, a concept with parallels to interlocking cogwheels in macroscopic motors. However, these molecular cogwheels in the chemotaxis machinery undergo shape changes during function, which may benefit from the symmetry mismatch of MotA/B. One possibility is that the open MotA/B clefts[64,65] allow rapid binding or release of the torque helix without the need for a rate-limiting induced-fit process. Cleft closure would grasp the FliG$_{D5}$ torque helix tightly. (**b**) Complementary electrostatics and sterics of the torque helix of FliG$_{D5}$ and the MotA/B stator. An electrostatic surface representation of the FliG$_{D5}$ domain shows that the torque helix is presented as an isolated feature and is negatively charged. (**c**) A schematic mechanism for bidirectional MotA/B-dependent rotation of the C-ring by moving the MotA/B binding site from the outside to the inside of the ring.

# Reporting Summary

## Statistics

For all statistical analyses, confirm that the following items are present in the figure legend, table legend, main text, or Methods section.

| n/a | Confirmed | |
|---|---|---|
| ☐ | ☒ | The exact sample size (*n*) for each experimental group/condition, given as a discrete number and unit of measurement |
| ☒ | ☐ | A statement on whether measurements were taken from distinct samples or whether the same sample was measured repeatedly |
| ☒ | ☐ | The statistical test(s) used AND whether they are one- or two-sided<br>*Only common tests should be described solely by name; describe more complex techniques in the Methods section.* |
| ☒ | ☐ | A description of all covariates tested |
| ☒ | ☐ | A description of any assumptions or corrections, such as tests of normality and adjustment for multiple comparisons |
| ☒ | ☐ | A full description of the statistical parameters including central tendency (e.g. means) or other basic estimates (e.g. regression coefficient) AND variation (e.g. standard deviation) or associated estimates of uncertainty (e.g. confidence intervals) |
| ☒ | ☐ | For null hypothesis testing, the test statistic (e.g. *F*, *t*, *r*) with confidence intervals, effect sizes, degrees of freedom and *P* value noted<br>*Give P values as exact values whenever suitable.* |
| ☒ | ☐ | For Bayesian analysis, information on the choice of priors and Markov chain Monte Carlo settings |
| ☒ | ☐ | For hierarchical and complex designs, identification of the appropriate level for tests and full reporting of outcomes |
| ☒ | ☐ | Estimates of effect sizes (e.g. Cohen's *d*, Pearson's *r*), indicating how they were calculated |

*Our web collection on statistics for biologists contains articles on many of the points above.*

## Software and code

Policy information about availability of computer code

| | |
|---|---|
| Data collection | EPU version 3.0.0.4164 was used for the CCW dataset. EPU version 3.5.1.6034 was used for both CW datasets. |
| Data analysis | For map refinement CryoSPARC 4.2.1 was used,<br>For model building Phenix 1.20.1-4, Coot 0.9.8.8, MolProbity, UCSF Chimera 1.17.2, AlphaFold 2.3.2 was used. For illustrations and animations UCSF ChimeraX 1.7, Adobe Ilustrator 27.7, Blender 3.5, Molecular Node 2.8 was used. |

For manuscripts utilizing custom algorithms or software that are central to the research but not yet described in published literature, software must be made available to editors and reviewers. We strongly encourage code deposition in a community repository (e.g. GitHub). See the Nature Portfolio guidelines for submitting code & software for further information.

## Data

Policy information about availability of data

All manuscripts must include a data availability statement. This statement should provide the following information, where applicable:
- Accession codes, unique identifiers, or web links for publicly available datasets
- A description of any restrictions on data availability
- For clinical datasets or third party data, please ensure that the statement adheres to our policy

Source data (uncropped micrograph used for Figure 1b) are available with this publication. All raw, processed, and interpreted data that support the findings of this study are available in public repositories. Raw micrographs have been deposited with EMPIAR (https://www.ebi.ac.uk/empiar/) and accession codes EMPIAR-11597,

EMPIAR-11891, and EMPIAR-11892. CryoEM maps have been deposited at the EMDB (https://www.ebi.ac.uk/emdb/) with the accession codes EMD-41100, EMD-41101, EMD-41102, EMD-41103, EMD-41104, EMD-43256, EMD-43258, EMD-43327, and EMD-43328. Atomic coordinates of the 34-mer CCW C-ring and the 33-mer MS-ring have been deposited at the Protein Data Bank (www.rcsb.org) with the accession codes rcsb.org/structure/8t8o and rcsb.org/structure/8t8p. Atomic coordinates for the single subunit of the isolated CW-locked C-ring are deposited with the accession code rcsb.org/structure/8vib, the 34-mer isolated CW-locked C-ring are deposited with accession code rcsb.org/structure/8vid. Coordinates for a single subunit of the CW-locked C-ring bound to a partner protein have the accession code rcsb.org/structure/8vkq, and the 34-mer of the CW-locked C-ring bound to a partner protein have the accession code rcsb.org/structure/8vkr. Previously reported structures or computational models used to support this work are: Thermotoga maritima FliG (rcsb.org/structure/1lkv30; rcsb.org/structure/5tdy34), Helicobacter pylori FliG (rcsb.org/structure/3usw31, rcsb.org/structure/4fq032), T. maritima FliN (rcsb.org/structure/1yab42), S. enterica FliM:FliN fusion (rcsb.org/structure/4yxb39), S. enterica flagellar basal body (rcsb.org/structure/7cgo12), A. aeolicus MotA (rcsb.org/structure/8gqy71), C. jejunji MotA/B (rcsb.org/structure/6ykm65), C. sporogenes MotA/B (rcsb.org/structure/6ysf64), B. subtilis MotA/B (rcsb.org/structure/6ysl64), E. coli YcgR (rcsb.org/structure/5y6h56), T. maritima CheY-FliM1-16 (rcsb.org/structure/4iga57), E. coli quinol:fumarate reductase (rcsb.org/structure/1kf658), and FliO (alphafold.ebi.ac.uk/entry/A0A5C2LXN829).

## Research involving human participants, their data, or biological material

Policy information about studies with human participants or human data. See also policy information about sex, gender (identity/presentation), and sexual orientation and race, ethnicity and racism.

| | |
|---|---|
| Reporting on sex and gender | No human research participants were involved. |
| Reporting on race, ethnicity, or other socially relevant groupings | No human research participants were involved. |
| Population characteristics | No human research participants were involved. |
| Recruitment | No human research participants were involved. |
| Ethics oversight | No human research participants were involved. |

Note that full information on the approval of the study protocol must also be provided in the manuscript.

# Field-specific reporting

Please select the one below that is the best fit for your research. If you are not sure, read the appropriate sections before making your selection.

☒ Life sciences    ☐ Behavioural & social sciences    ☐ Ecological, evolutionary & environmental sciences

For a reference copy of the document with all sections, see nature.com/documents/nr-reporting-summary-flat.pdf

# Life sciences study design

All studies must disclose on these points even when the disclosure is negative.

| | |
|---|---|
| Sample size | For the wild-type FliFGMN in the CCW pose, 295,031 particles from 34,381 movies were used. For the CW-locked FliFGMN, 43,741 particles from 35,552 movies were used. For the CW-locked FliFGMN bound to a regulator, 59,404 particles from 26,130 movies were used. |
| Data exclusions | For all datasets, CTF fit above 10Å was excluded as they did not provide high resolution information. For wild-type FliFGMN in the CCW pose, in 2D classification all contaminant protein and low signal to noise ratio particles were removed, in heterogeneous refinement any remaining particles that was not a MSC-ring was removed by 3D alignment and in final refinement of the C-ring, MS-rings was removed and for the MS-ring, the C-ring particles were removed by particle subtraction. For FliFGMN in the CW pose, all contaminant protein and low signal to noise ratio particles were removed in 2D classification and heterogeneous refinement, and the final refinement of the C-ring was performed by masking three subunits of the C-ring to obtain high resolution. For FliFGMN in the CW pose with a regulator, all contaminant protein and low signal to noise ratio particles were removed in 2D classification and heterogeneous refinement, and the final refinement of the C-ring was performed by masking three subunits of the C-ring to obtain high resolution. |
| Replication | For the wild-type FliFGMN in the CCW pose, 295,031 protein complexes from one prepared grid was used and the calculations were performed two independent times. For the CW-locked FliFGMN, 43,741 protein complexes from one prepared grid was used and the calculations were performed once. For the CW-locked FliFGMN bound to a regulator, 59,404 protein complexes from one prepared grid was used and the calculations were performed once. |
| Randomization | Randomization was not performed. |
| Blinding | Blinding was not performed. |

# Reporting for specific materials, systems and methods

We require information from authors about some types of materials, experimental systems and methods used in many studies. Here, indicate whether each material, system or method listed is relevant to your study. If you are not sure if a list item applies to your research, read the appropriate section before selecting a response.

## Materials & experimental systems

| n/a | Involved in the study |
|-----|----------------------|
| ☒ ☐ | Antibodies |
| ☒ ☐ | Eukaryotic cell lines |
| ☒ ☐ | Palaeontology and archaeology |
| ☒ ☐ | Animals and other organisms |
| ☒ ☐ | Clinical data |
| ☒ ☐ | Dual use research of concern |
| ☒ ☐ | Plants |

## Methods

| n/a | Involved in the study |
|-----|----------------------|
| ☒ ☐ | ChIP-seq |
| ☒ ☐ | Flow cytometry |
| ☒ ☐ | MRI-based neuroimaging |

## Plants

| Seed stocks | N/A |
|-------------|-----|
| Novel plant genotypes | N/A |
| Authentication | N/A |

