## [Peer Review File · Nature Microbiology]

Peer Review Information

Journal: Nature Microbiology

Manuscript Title: CryoEM structures reveal how the bacterial flagellum rotates and switches direction

Corresponding author name(s): Professor Tina Iverson

Editorial Notes:

This manuscript has been previously reviewed at another journal. This document only contains reviewer comments, rebuttal and decision letters for versions considered at Nature Microbiology. Mentions of prior referee reports have been redacted

Reviewer Comments & Decisions:

Decision Letter, initial version:

Message: 30th October 2023

Dear Professor Iverson,

Thank you for your patience while your manuscript "Structural basis for motor rotation and directional switching of bacterial flagella" was under peer review at Nature Microbiology. It has now been seen by our referees, whose expertise and comments you will find at the end of this email. In the light of their advice, we have decided that we cannot offer to publish your manuscript in Nature Microbiology.

From the reports, you will see that while they find your work of some potential interest, the referees raise concerns about whether sufficient data and structural analyses are provided to support some of the conclusions drawn over the CCW state of the C-ring. Unfortunately, we feel that these criticisms are sufficiently important as to preclude publication of your work in Nature Microbiology, and that to attempt to address them would take considerable time and effort, without guarantee that the data obtained would resolve the issues.

Although we cannot offer to publish your paper in Nature Microbiology, I have discussed your manuscript and the reviewers' comments with our colleagues at Nature Communications, and they are very interested. They would send an appropriately revised version back to reviewers if the manuscript is transferred to their journal. If you wish to have your revised paper considered by Nature Communications, please use the link to the Springer Nature manuscript transfer service in the footnote once the revision is ready, and include a point-by-point response to the reviewers' concerns. Your handling editor at Nature Communications would be Cesar Sanchez (cesar.sanchez@nature.com), please feel free to contact him if you have any questions. Please note that Nature Communications is a fully open access journal; for information about article processing charges, open access funding, and advice and support from Nature Research, please consult the Nature Communications Open Access page (nature.com/ncomms/about/open-access).

To transfer your manuscript please use our manuscript transfer portal. You will not have to re-supply manuscript metadata and files, unless you wish to make modifications. For more information, please see our manuscript transfer FAQ page.

I am sorry that we cannot be more positive on this occasion, but hope that you find the referees' comments helpful when preparing your paper for resubmission elsewhere.

Yours sincerely,

2Reviewer Expertise:

Referee #1: structural biology, chemotaxis, flagella

Referee #2: structural biology, molecular machines

Referee #3: structural biology, secretion systems

Reviewers Comments:

Reviewer #1 (Remarks to the Author):

The manuscript presents the structure of the flagellar C-ring (aka the switch). The structural data obtained provide elegant explanations for many of the previously obtained biochemical and genetic data, which appeared eventually conflicting. The authors do a great job in integrating these data with the structural information provided by this study. Most strikingly for me was to see the interweaving of subunits within the ring, which could not be anticipated otherwise. I congratulate the authors to this amazing piece of work and suggest publication after some minor edits:

-For non-expert readers, please add two sentences describing the complete architecture of the flagellum (as shown in Fig. 1a). They authors start very straight into the C-ring, assuming every reader knows the flagellar architecture by heart ;-)

-Main section: „...Bacterial chemotaxis is controlled by the flagellum, a multicomponent proteinaceous machine...“; Does the flagellum really control bacterial chemotaxis? I would say, the flagellum is an essential element of chemotaxis. Maybe rephrase this sentence

„...It is critical for the biogenesis of flagella and may be the first region to assemble1-3...“;
You might want consider the following reviews as well: PMID: 26195616, PMID: 23600726

Reviewer #2 (Remarks to the Author):

This nice manuscript from redacted authors describes the first near-atomic-resolution structure of the bacterial flagellar C ring based on a cryoEM dataset of recombinantly expressed and purified M, S and C-ring (FliFGMN) of *S. enterica* serovar motors using the long-established pKLR3 plasmid, with subsequent modelling using starting structures from AlphaFold. The paper describes MS and C-ring symmetry- and axis-mismatch; description of molecular organization within these structures; description of FliFGMN folds; and a description of inferred CW and CCW switch poses based on the model from *T. maritima* CW-stabilised FliG and binding partners/sites identified in other research.

This is a substantial advance, involving a range of image processing manoeuvres to attain

2the final medium-resolution structure. The work contributes a number of important structural insights into underlying mechanisms, including torque generation (by contributing the in situ structure of the five-domain FliG) and C ring assembly (by contributing the architecture of the FliM(1):FliN(3) region of the C ring to explain how helix is formed instead of stacked rings of SPOA domains), and interaction with other proteins. Because the study consists of a single structure with no mutagenesis and additional structure determination, interpretation of parts of the study remain more speculative, for example the possible binding locations of CheY, directional switching, and torque generation.

The results are a substantial step forward and will inform the work of many groups, answering a number of long-standing and significant questions. The writing is overall good, although suffers from substantial hyperbole which will be a distraction for non-expert readers in assessing the true importance of the work. Nevertheless, there are many points that need to be addressed to make this paper suitable for publication

Major points

- As you'll see in my notes below, the authors do not appropriately cite contemporary literature. For example, the introduction cites three 15-20 year-old reviews that are now very outdated. Please reference (and read!) contemporary reviews-there are plenty out there. In many cases you should be citing primary literature instead of reviews. Work of other groups is often not given appropriate respect throughout the manuscript, which is sad given the amount of work performed by many other groups. Please acknowledge this hard work - your work is a huge step forward, but do pay due respect to all of this previous work. It doesn't diminish from your contributions.
- I have a concern about the physiological relevance of the reported 33-fold symmetry of the MS ring. Kawamoto et al have made a compelling argument that the native symmetry is 34 (cited but, as per my previous point, but not properly addressed in your manuscript - 'Native Flagellar MS Ring Is Formed by 34 Subunits with 23-Fold and 11-Fold Subsymmetries'. Nature Communications 12, no. 1 (9 July 2021): 4223. <https://doi.org/10.1038/s41467-021-24507-9>.) The pKLR3 plasmid from Khan lacks the important components such as FliPQR, FliH, all of which make interactions with, and therefore quite possibly template correct MS-ring symmetry. (pKLR3 does contain full-length FliF, and it is critical that this be explicitly highlighted, as truncated FliF, used in many studies, is responsible for deviations from 34-fold symmetry!). Studies by the Lea are also important. Why do the authors get a different result? Its ok to say you don't know!
- I have some concerns over the lack of validation of hypotheses. While I'm wary of recommending the authors perform any additional experiments, a substantial portion of the paper consists of hypotheses about CW/CCW switching, torque generation, etc, with no substantiation beyond "this is consistent with previous mutants" (it's easy to subconsciously dismiss mutants that are inconsistent, no matter how rigorous you are). The writing style avoids making direct statements about what an updated model for rotation and switching would look like, whilst vaguely dismissing existing research without detailed critical phrasing, particularly so in the discussion section. A little more humility would communicate that these are just models more effectively. As above - your contribution is already substantial; overstating the novelty or robustness of models merely undermines your readers' faith in you.

3- How do the authors reconcile their insights with Liu's work suggesting large scale movement of the N-terminus of FlIG? Why resort to suggesting MotAB moves in the video? How would CheY power that? Be concrete.

Minor points

- Please always add line numbers to submitted manuscripts.
- "the stability and flexibility necessary for rapid rotation" - physics at nanoscale behaves non-intuitively for us macro-scale organisms, and this sentence is sufficiently vague to inadequately describe exactly what the authors are referring to. Thermal bombardment and ultra-low Reynolds numbers are the problems encountered at nanoscale, and it's worth joining-the-dots as to how this unsubstantiated 'stability and flexibility' best tackles these problems during rapid rotation. Consider removing.
- "serving as a type III secretion system": the export apparatus IS a type III secretion system.
- "High-resolution structures of the rod, export apparatus, MS-ring, P-ring, L-ring, flagellar hook, and flagellar filament provided insight into the organization of these flagellar components and suggest how they withstand rapid flagellar rotation": As with my previous comment, I don't think these studies addressed "how they withstand rapid flagellar rotation". Consider removing, or substantiating clearly and specifically.
- "has resisted all efforts at structure determination": not true - what of DeRosier's structures? Liu's subtomogram average structures to sufficient resolution to discern subunits? Many crystal structures of individual units? Just because these weren't at sufficient resolution for molecular modelling doesn't mean they were vital pieces of the narrative. Also address subsequent allusions to this.
- "may be the first region to assemble": Not true; it's almost certain that FlIPQR/FlhB and FlIF come first (and thus the 33 FlIF may be an artefact). References 1-3 are inappropriate references for this claim. Please familiarise yourself with the wealth of studies on assembly order of both flagella and injectisomes. Also address subsequent allusions to this.
- Be careful with adjectives - e.g., "demanding": leave it up to your readers as to whether it is subjectively "demanding". Can a molecule have a demanding task?
- Paragraph commencing "The switch has posed a challenge for structural studies.": this fails to recognise that this data are not all consistent, ie., someone is wrong, or we're missing something important - this is nicely discussed in the "Flagellar self-assembly" section of Beeby et al's "The Convergent Evolution of Archaeella, Flagella and Cilia" FEMS Microbiology Reviews 44, no. 3 (1 May 2020): 253–304. <https://doi.org/10.1093/femsre/fuaa006>. (this review could serve well in place of the outdated reviews currently cited).
- "employ particle subtraction at the level of the micrograph" is far too vague. What did you subtract? The details are in M&M but should be described clearly and concisely in the Results so your readers don't need to flip between sections of the manuscript.
- The authors claim a modified cryoEM computational workflow, but it doesn't seem particularly novel. All approaches have become fairly routine in recent years. Suggest rephrasing.
- "Consistent with an adaptive stoichiometry": it isn't inconsistent, but this wording is misleading, and it certainly doesn't "support" let alone "confirm" (which "consistent" alludes to). The results on adaptive stoichiometry come exclusively from fluorescent tag results, which swim poorly. We cannot rule out that the tag interferes with C ring architecture. Suggest removing this sentence to avoid opening that can of worms.
- AlphaFold2 plots should be included for models so readers can assess their confidence.

4- Citing cryoSPARC three times in the same sentence is unnecessary.
- Figure 2, which provides information on the 'layered' conformation of the C-ring, would benefit from showing a few conformations of just one 'stave' next to the ring?
- "however, when FliG is assembled into the switch, it clearly contains five domains": I'd suggest bringing forward mention of Extended Fig. 6 to here, and consider making the extended figure a part of Fig. 3?
- The conventional view of models of FliG in the C-ring has been cross-sections through the C-ring, making Fig. 3 difficult to interpret. Suggest including a version of Fig 5C in Fig 3 to assist reader understanding.
- "Given the uniqueness of the cryoEM structure, we leveraged published mutagenesis and thiol crosslinking to validate the assembled switch" - this is not good science! It's difficult to substantiate how you avoid cherry-picking results, so you are "consistent" with the data, but I'd dispute that you're "validating". Please be careful with such coaxing language which unfairly directs the attention of beginners to the field.
- The helical structure of the base of the C ring has previously been proposed (almost ten years ago - but due to the outdated reviews cited, the authors are not aware, as they don't even cite the paper.) Please accord due respect to this previous work: McDowell, et al. 'Characterisation of Shigella Spa33 and Thermotoga FliM/N Reveals a New Model for C-Ring Assembly in T3SS'. Molecular Microbiology 99, no. 4 (1 February 2016): 749–66. <https://doi.org/10.1111/mmi.13267>.
- Fig. 5e: Why show a hexamer in the 'circular' cartoon? Why not make it an arc from a 34-repeat structure? I find the hexamer cartoon confusing.
- "This proposal leverages past findings suggesting that MotA/B interacts": this is crucial information but the correct citations are not acknowledged.
- Supplementary movie 4: why does the MS ring rotate faster than the C ring when rotating CW?
- "Non-coaxial stacking is also observed between MS-ring and the basal body": the MS ring is a part of the basal body! What are you referring to? Be explicit.
- "Taken together, we propose that the MS-ring may wobble during flagellar rotation (Supplementary Movie 5) which could prevent an energetic minimum during flagellar rotation": Why? Spell it out - it's not clear to me.
- "Finally, the structure suggests uniting themes": what are you referring to? Be clear, concise, and concrete.

Reviewer #3 (Remarks to the Author):

This manuscript describes cryo-EM structures of the flagellar motor from Salmonella at ~4Å resolution. Perhaps the most significant finding is that the C-ring is composed of 34-mer of FliG:FliM:3FliN. The manuscript is technically sound and cryo-EM structures are excellently presented. The C-ring model appears to be consistent with many published mutagenesis and crosslinking data. However, the manuscript is poorly written, difficult to read, and full of speculations.

Major comments:

1. Symmetry mismatch between FliG and FliF should be thoroughly examined. It is not

5clear if the symmetry mismatch is physiologically relevant given the complicated sample preparation procedure involved and heterogeneous particles as shown in the cryo-EM image (Fig. 1b).

2. It is also not clear if the 4-degree tilt between the MS-ring and the C-ring is relevant to the function of the motor.

3. The cryo-EM structures are resulted from an overexpressed complex that was assumed to be in CCW conformation. However, there is no evidence to support this model.

4. Moreover, the manuscript did not provide any data to support the "CW" model.

***Nature Communications* is the Nature Portfolio flagship Open Access journal. If you would like this work to be considered for publication there, you can easily transfer the manuscript by following the instructions below. It is not necessary to reformat your paper. Once all files are received, the editors at *Nature Communications* will assess your manuscript's suitability for potential publication; they aim to provide feedback quickly, with a median decision time of 8 days for first editorial decisions on suitability. Since your paper has been peer reviewed at this journal, the referee reports will also be transferred and assessed by the editorial team. In some cases, papers are accepted without further peer review, providing a rapid path to publication. The journal is also proud to offer double blind and transparent peer review options. For 2021, the 2-year impact factor for *Nature Communications* is 17.694 and the 2-year median is 10 (for further information on journal impact factors, please visit our Nature journals metrics page). Our open access pages contain information about article processing charges, open access funding, and advice and support from Springer Nature.

** Although we cannot offer to publish your manuscript, we believe the editors at our sister journal, *Communications Biology*, will find it interesting and recommend you transfer it there.

Communications Biology is a selective Nature Portfolio title publishing Open Access research that brings new insight in all areas of the biological sciences. Additional journal metrics and information can be found here. Their editors prioritise good author service, fast peer review (in 2021, the median time to decision after first review was 40 days), and are happy to answer any questions you may have (commsbio@nature.com). The journal has an Impact Factor of 6.548, a CiteScore of 6.0 and a Scimago quartile ranking of Q1.

Please note that *Communications Biology* is a fully open-access journal and an article processing charge will apply to any papers accepted for publication. Our open access pages contain information about article processing charges, open access funding, and advice and support from Springer Nature.

If you wish to transfer your manuscript to *Communications Biology*, please use our manuscript transfer portal using the link below to initiate the transfer to this journal (or to another journal of your choice in the Nature Research portfolio). If you transfer to Nature-

6branded journals or to the Communications journals, you will not have to re-supply manuscript metadata and files. This link can only be used once and remains active until used. For more information, please see our manuscript transfer FAQ page.

***Scientific Reports* publishes primary research from all areas of the natural and clinical sciences that is judged to be scientifically valid and technically sound, whatever the considered significance. If you would like this work to be considered for publication in *Scientific Reports*, you can easily transfer the manuscript by following the instructions below. Your manuscript will be handled by an academic scientist who is an Editorial Board Member and will manage the peer review process and decide whether a paper should be accepted for publication. Most submissions are peer reviewed by one or more referees as well as the editorial board member and you can expect to receive an editorial decision within 56 days. Over 55% of the papers are published following peer review. To discover more about this journal and, should you wish, have your paper considered the Editorial Board of *Scientific Reports*, please use the link to the manuscript transfer service provided in the footnote below. Please see our open access pages for information about article processing charges, open access funding, and advice and support from Springer Nature.

Author Rebuttal to Initial comments

Reviewer #1 (Remarks to the Author):

The manuscript presents the structure of the flagellar C-ring (aka the switch). The structural data obtained provide elegant explanations for many of the previously obtained biochemical and genetic data, which appeared eventually conflicting. The authors do a great job in integrating these data with the structural information provided by this study. Most strikingly for me was to see the interweaving of subunits within the ring, which could not be anticipated otherwise. I congratulate the authors to this amazing piece of work and suggest publication after some minor edits:

-For non-expert readers, please add two sentences describing the complete architecture of the flagellum (as shown in Fig. 1a). They authors start very straight into the C-ring, assuming every reader knows the flagellar architecture by heart ;-)

→ Response: We thank the reviewer for the support of this work, in the revised version, we expanded the first paragraph to better explain the architecture of the flagellar motor.

-Main section: „...Bacterial chemotaxis is controlled by the flagellum, a multicomponent proteinaceous machine...”; Does the flagellum really control bacterial chemotaxis? I would say, the flagellum is an essential element of chemotaxis. Maybe rephrase this sentence.

→ Response: Rephrased as suggested.

-“...It is critical for the biogenesis of flagella and may be the first region to assemble1-3...”; You might want consider the following reviews as well: PMID: 26195616, PMID: 23600726

→ Response: We modified the text for improved accuracy as relates to assembly and we cited the suggested reviews.Reviewer #2 (Remarks to the Author):

This nice manuscript from redacted authors describes the first near-atomic-resolution structure of the bacterial flagellar C ring based on a cryoEM dataset of recombinantly expressed and purified M, S and C-ring (FliFGMN) of *S. enterica* serovar motors using the long-established pKLR3 plasmid, with subsequent modelling using starting structures from AlphaFold. The paper describes MS and C-ring symmetry- and axis-mismatch; description of molecular organization within these structures; description of FliFGMN folds; and a description of inferred CW and CCW switch poses based on the model from *T. maritima* CW-stabilised FliG and binding partners/sites identified in other research.

This is a substantial advance, involving a range of image processing manoeuvres to attain the final medium-resolution structure. The work contributes a number of important structural insights into underlying mechanisms, including torque generation (by contributing the in situ structure of the five-domain FliG) and C ring assembly (by contributing the architecture of the FliM(1):FliN(3) region of the C ring to explain how helix is formed instead of stacked rings of SPOA domains), and interaction with other proteins. Because the study consists of a single structure with no mutagenesis and additional structure determination, interpretation of parts of the study remain more speculative, for example the possible binding locations of CheY, directional switching, and torque generation.

The results are a substantial step forward and will inform the work of many groups, answering a number of long-standing and significant questions. The writing is overall good, although suffers from substantial hyperbole which will be a distraction for non-expert readers in assessing the true importance of the work. Nevertheless, there are many points that need to be addressed to make this paper suitable for publication.

→ Response: We thank the reviewer for the support of this work and for calling our intention to areas for improvement.

In terms of the comment that some of the additional states are speculative, our revision includes two additional experimental structures, which support our conclusions. These are the CW-locked state and the CW-locked state bound to a partner protein. In the revised text, we increased the focus on these experimental structures and removed all computational docking.

Major points

- As you'll see in my notes below, the authors do not appropriately cite contemporary literature. For example, the introduction cites three 15-20 year-old reviews that are now very outdated. Please reference (and read!) contemporary reviews-there are plenty out there. In many cases, you should be citing primary literature instead of reviews. Work of other groups is often not given appropriate respect throughout the manuscript, which is sad given the amount of work performed by many other groups.

Please acknowledge this hard work - your work is a huge step forward, but do pay due respect to all of this previous work. It doesn't diminish from your contributions.

→ Response: This point was well taken into consideration in the revised manuscript. We wish to assure the reviewer that the omission of citations for some primary literature was not due to a lack of respect for other groups or unawareness of the literature. (Among our authors are individuals with nearly 50 years of experience in the flagellar motor and bacterial chemotaxis.) It was due to a strict limitation of this journal on the total number of references allowed. This problem is improved in the revised manuscript thanks to the editor, who increased the maximal number of references given the reviewer's interest in additional citations. With respect to the comment about the reviews, we selected three reviews from different scientific lineages to attempt to represent schools of thought in the field. We cited these for central aspects of background that were not the main emphasis of the paper or where we were confirming dogma. Given this concern, we retained the 2020 review but replaced the two older reviews with more recent ones.

- I have a concern about the physiological relevance of the reported 33-fold symmetry of the MS ring. Kawamoto et al have made a compelling argument that the native symmetry is 34 (cited but, as per my previous point, but not properly addressed in your manuscript - 'Native Flagellar MS Ring Is Formed by 34 Subunits with 23-Fold and 11-Fold Subsymmetries'. Nature Communications 12, no. 1 (9 July 2021): 4223. <https://doi.org/10.1038/s41467-021-24507-9>). The pKLR3 plasmid from Khan lacks the important components such as FliPQR, FliH, all of which make interactions with, and therefore quite possibly template correct MS-ring symmetry. (pKLR3 does contain full-length FliF, and it is critical that this be explicitly highlighted, as truncated FliF, used in many studies, is responsible for deviations from 34-fold symmetry!). Studies by the Lea are also important. Why do the authors get a different result? Its ok to say you don't know!

→ Response: The stoichiometry of the MS-ring is an important point that has been extensively debated in recent literature.

In the combined MS- and C-ring structure reported here, we are confident that the MS-ring is a 33-mer when the C-ring is a 34-mer under the conditions of our sample preparation. In the revised version, we added a section to the discussion describing the existing controversy over MS-ring stoichiometry. We explicitly indicate that the MS-ring controversy is not resolved by our data.

- I have some concerns over the lack of validation of hypotheses. While I'm wary of recommending the authors perform any additional experiments, a substantial portion of the paper consists of hypotheses about CW/CCW switching, torque generation, etc, with no substantiation beyond "this is consistent with previous mutants" (it's easy to subconsciously dismiss mutants that are inconsistent, no matter how rigorous you are). The writing style avoids making direct statements about what an updated model for rotation and switching would look like, whilst vaguely dismissing existing research without detailed

10critical phrasing, particularly so in the discussion section. A little more humility would communicate that these are just models more effectively. As above - your contribution is already substantial; overstating the novelty or robustness of models merely undermines your readers' faith in you.

→ Response: In response to the concern about how well the data support CCW/CW switching, we provide two additional experimental structures: the CW state and the CW state bound to a stabilizing protein. We removed the computational docking of quinol:fumarate reductase and CheY. We only retained a subset of the information on MotA/B to provide context for the mechanism.

In terms of using published mutational analysis to validate the structure, we analyzed the mutations in aggregate rather than individually, and no known mutations were omitted from the analysis (Fig. 4b, Extended Data Figure 4). This included >20 assembly-deficient mutants that are consistent with our structure and > 100 directionally biasing mutants that are consistent with our proposed switching mechanism. In aggregate, we find that both sets of mutations are overwhelmingly consistent with our structure and proposed mechanisms. In addition, individual domains also match crystal structures.

We apologize that the writing tone of the original manuscript gave the reviewer an impression of a lack of humility. It stemmed from being extremely excited about how this work advances what we know about chemotaxis. For this reason, we also focused on aspects of the work that break new ground. In the revision, we adjusted the content and language throughout the revised manuscript. We now indicate additional aspects that confirm past publications, such as the structure of the MS-ring, the spiral of FliM and FliN, and the binding site for MotA/B.

- How do the authors reconcile their insights with Liu's work suggesting large scale movement of the N-terminus of FliG? Why resort to suggesting MotAB moves in the video? How would CheY power that? Be concrete.

→ Response: In the revised version, we now include the experimental structure of a CW-locked mutant and indeed find a 180° rotation in N-terminus of FliG that opens a binding cleft between the N- and C-terminal domains. We also now cite the primary literature demonstrating that MotA/B changes from the outside of the ring during CCW rotation to the inside of the ring during CW rotation (Chang and Liu (2019) eLife).

Minor points

- Please always add line numbers to submitted manuscripts.

→ Response: We added line numbers to the revised version.

- "the stability and flexibility necessary for rapid rotation" - physics at nanoscale behaves non-intuitively for us macro-scale organisms, and this sentence is sufficiently vague to inadequately describe exactly

what the authors are referring to. Thermal bombardment and ultra-low Reynolds numbers are the problems encountered at nanoscale, and it's worth joining-the-dots as to how this unsubstantiated 'stability and flexibility' best tackles these problems during rapid rotation. Consider removing.

→ Response: We modified the phrasing to improve the clarity.

- "serving as a type III secretion system": the export apparatus IS a type III secretion system.

→ Response: Corrected

- "High-resolution structures of the rod, export apparatus, MS-ring, P-ring, L-ring, flagellar hook, and flagellar filament provided insight into the organization of these flagellar components and suggest how they withstand rapid flagellar rotation": As with my previous comment, I don't think these studies addressed "how they withstand rapid flagellar rotation". Consider removing, or substantiating clearly and specifically.

→ Response: We modified the phrasing to reflect the extensive body of literature showing that domain swaps increase the stability of protein multimers.

- "has resisted all efforts at structure determination": not true - what of DeRosier's structures? Liu's subtomogram average structures to sufficient resolution to discern subunits? Many crystal structures of individual units? Just because these weren't at sufficient resolution for molecular modelling doesn't mean they were vital pieces of the narrative. Also address subsequent allusions to this.

→ Response: The sentence was meant to state that high-resolution structures of the assembled C-ring have not been reported. We thank you for bringing this error to our attention and have modified it for accuracy. The additional studies containing tomograms and crystal structures were cited throughout the paper. A citation to the structure from David's group has now been added to the revised version.

It is worth noting that our results indicate that the resolution of past cryoEM envelopes was NOT the barrier to modeling. Instead, individual crystal structures of isolated domains had non-physiological domain swaps in the unassembled and isolated proteins. This is illustrated in Extended Data Fig. 4.

- "may be the first region to assemble": Not true; it's almost certain that FlIPQR/FlhB and FlIF come first (and thus the 33 FlIF may be an artefact). References 1-3 are inappropriate references for this claim. Please familiarise yourself with the wealth of studies on assembly order of both flagella and injectisomes. Also address subsequent allusions to this.

→ Response: Corrected as pointed out. We additionally now cite two recent reviews on flagellar assembly.

- Be careful with adjectives - e.g., "demanding": leave it up to your readers as to whether it is subjectively "demanding". Can a molecule have a demanding task?

→ Response: Changed as suggested

- Paragraph commencing "The switch has posed a challenge for structural studies.": this fails to recognise that this data are not all consistent, ie., someone is wrong, or we're missing something important - this is nicely discussed in the "Flagellar self-assembly" section of Beeby et al's "The Convergent Evolution of Archaeella, Flagella and Cilia" FEMS Microbiology Reviews 44, no. 3 (1 May 2020): 253–304. <https://doi.org/10.1093/femsre/fuaa006>. (this review could serve well in place of the outdated reviews currently cited).

→ Response: We removed most of this paragraph and cited this review, as requested.

- "employ particle subtraction at the level of the micrograph" is far too vague. What did you subtract? The details are in M&M but should be described clearly and concisely in the Results so your readers don't need to flip between sections of the manuscript.

→ Response: We added a description of particle subtraction to the beginning of the results section (lines 75 – 87).

- The authors claim a modified cryoEM computational workflow, but it doesn't seem particularly novel. All approaches have become fairly routine in recent years. Suggest rephrasing.

→ Response: The novel modification of the workflow was the particle subtraction technique. The modified text (lines 75 – 87) now clarifies why particle subtraction differs from the standard cryoEM workflow.

- "Consistent with an adaptive stoichiometry": it isn't inconsistent, but this wording is misleading, and it certainly doesn't "support" let alone "confirm" (which "consistent" alludes to). The results on adaptive stoichiometry come exclusively from fluorescent tag results, which swim poorly. We cannot rule out that the tag interferes with C ring architecture. Suggest removing this sentence to avoid opening that can of worms.

→ Response: We modified the text.

- AlphaFold2 plots should be included for models so readers can assess their confidence.

→ Response: AlphaFold2 plots are now added in Extended Data Figure 2.

- Citing cryoSPARC three times in the same sentence is unnecessary.

→ Response: Corrected.

- Figure 2, which provides information on the 'layered' conformation of the C-ring, would benefit from showing a few conformations of just one 'stave' next to the ring?

- "however, when FliG is assembled into the switch, it clearly contains five domains": I'd suggest bringing forward mention of Extended Fig. 6 to here, and consider making the extended figure a part of Fig. 3?

- The conventional view of models of FliG in the C-ring has been cross-sections through the C-ring, making Fig. 3 difficult to interpret. Suggest including a version of Fig 5C in Fig 3 to assist reader understanding.

→ Response to the three above points: As suggested, we repaneled some of the figures.

- "Given the uniqueness of the cryoEM structure, we leveraged published mutagenesis and thiol crosslinking to validate the assembled switch" - this is not good science! It's difficult to substantiate how you avoid cherry-picking results, so you are "consistent" with the data, but I'd dispute that you're "validating". Please be careful with such coaxing language which unfairly directs the attention of beginners to the field.

→ Response: Mutagenesis is the predominant method used to validate cryoEM structures. The large body of available mutants means that designing validation mutations was unnecessary. Instead, we analyzed the mutations in aggregate rather than individually, which avoids cherry picking. No known mutations were omitted from the analysis (Extended Data Figures 3, 8). This included >20 assembly-deficient mutants that are consistent with our structure and > 100 directionally biasing mutants that are consistent with our proposed switching mechanism. Furthermore, these earlier studies were not biased, as they were not influenced by the structure of our current study. Notably, we find that, in aggregate, both sets of mutations are overwhelmingly consistent with our structure and proposed mechanisms. We clarified these points in the revised manuscript.

- The helical structure of the base of the C ring has previously been proposed (almost ten years ago - but due to the outdated reviews cited, the authors are not aware, as they don't even cite the paper.) Please accord due respect to this previous work: McDowell, et al. 'Characterisation of Shigella Spa33 and Thermotoga FliM/N Reveals a New Model for C-Ring Assembly in T3SS'. *Molecular Microbiology* 99, no. 4 (1 February 2016): 749–66. <https://doi.org/10.1111/mmi.13267>.

→ Response: We are well aware of past proposals in the literature for this spiral structure. As explained above, due to length limits and the strict limit on the number of allowed citations we found it necessary to focus on novel findings and cite reviews for general background. It should be noted that McDowell (2016) is not the first report of a helical structure at the base of the C-ring (and that we are quite familiar with this work). The first report was by Daniela Stock's group in 2010 (Lee et al (2010) *Nature*, see SI Figure 11d). This ground-breaking paper, which we did reference, describes the base of the ring as a 'spiral' and proposes a heterotetramer of FliN and FliM_C. Because this spiral has become an accepted model in the field, the section describing it was brief. We now expanded this and added the McDowell citation.

- Fig. 5e: Why show a hexamer in the 'circular' cartoon? Why not make it an arc from a 34-repeat structure? I find the hexamer cartoon confusing.

→ Response: In the revision, we modified the figure to be a 34-mer and used an inset to show how the individual blocks affect curvature.

- "This proposal leverages past findings suggesting that MotA/B interacts": this is crucial information but the correct citations are not acknowledged.
→ Response: Citation to Chang and Liu (2019) eLife is now added.
- Supplementary movie 4: why does the MS ring rotate faster than the C ring when rotating CW?
→ Response: This is explained by considering how two interacting cogwheels of different numbers of teeth interact. The smaller one will rotate more rapidly than the larger one. As MotA/B contains 5 binding sites for FliG torque helices, and the C-ring has 34 subunits, one rotation of the C-ring will involve ~7 rotations of MotA/B.
- "Non-coaxial stacking is also observed between MS-ring and the basal body": the MS ring is a part of the basal body! What are you referring to? Be explicit.
→ Response: We thank the reviewer for catching this typo. The sentence was meant to say between the MS-ring and the remainder of the basal body. This is now corrected.
- "Taken together, we propose that the MS-ring may wobble during flagellar rotation (Supplementary Movie 5) which could prevent an energetic minimum during flagellar rotation": Why? Spell it out - it's not clear to me.
→ Response: We expanded this section for clarity (lines 420 - 448). In short, the MS-ring is non coaxial with both C-ring and the remainder of the basal body. If there is axial rotation of the C-ring and rod during flagellar rotation, this would result in non-axial rotation of the MS-ring. Non-axial rotation would make this region of the motor appear to wobble when viewed edge on.
- "Finally, the structure suggests uniting themes": what are you referring to? Be clear, concise, and concrete.
→ Response: For clarity, this concluding paragraph is now modified.

Reviewer #3 (Remarks to the Author):

This manuscript describes cryo-EM structures of the flagellar motor from *Salmonella* at ~4Å resolution. Perhaps the most significant finding is that the C-ring is composed of 34-mer of FliG:FliM:3FliN. The manuscript is technically sound and cryo-EM structures are excellently presented. The C-ring model appears to be consistent with many published mutagenesis and crosslinking data. However, the manuscript is poorly written, difficult to read, and full of speculations.

→ Response: We improved the writing, and moved many of the details that are of interest to specialists to figure legends. We also moved all mechanisms and interpretations to the discussion section. To further decrease the speculations, we now include two additional experimental structures to support the CW state. In addition, we refocus the text on these experimental structures and remove computational docking of quinol:fumarate reductase and MotA/B.

Major comments:

1. Symmetry mismatch between FliG and FliF should be thoroughly examined. It is not clear if the symmetry mismatch is physiologically relevant given the complicated sample preparation procedure involved and heterogeneous particles as shown in the cryo-EM image (Fig. 1b).

→ Response: We agree with the reviewer that being certain of the symmetry mismatch is an important point. Our protein sample preparation protocol follows the standard in the field for the *Salmonella* MS- and C-rings and has been used in the field for over 20 years. Because the symmetry mismatch between the MS- and C-rings was unexpected, we wanted to be sure that this assignment was correct. To do this, we performed extensive subset classification on C-rings with 34-mer stoichiometry. This identified that 34-mer C-rings were only associated with 33-mer MS-rings in these samples under the conditions that we used for preparation. In the revised version, we extended the discussion of stoichiometry assignment (lines 560-575) to clarify this point.

In Figure 1b, the particles in the image have a range of orientations including en face views, tilted views, and side views. Because the orientation of the tilted views makes it appear as if some of the samples have differences in diameter, this could lead to the conclusion that the samples are heterogeneous. In the revised manuscript, we labeled a subset of the particles with their orientations so that this is not interpreted as heterogeneity, and we modified the legend to provide this clarification.

2. It is also not clear if the 4-degree tilt between the MS-ring and the C-ring is relevant to the function of the motor.

→ Response: The 4° tilt is clearly observed in the data of the CCW motor, is expected for a symmetry mismatch, is consistent with the previously reported tilt between the MS-ring and the LP-ring, and explains why the MS-ring looks 'thicker' on edge in low resolution structures of the combined MS- and C-rings than expected from the high-resolution structures. The new CW structures suggest that the

ability to tilt may facilitate transitions between the CCW and CW states, and this is now added to the discussion. We have now added this to the discussion in lines 420 – 448.

3. The cryo-EM structures are resulted from an overexpressed complex that was assumed to be in CCW conformation. However, there is no evidence to support this model.

→ Response: The isolated wild-type flagellar motor is known to purify as >99% CCW conformation and the purified protein only adopts the CW pose with mutation or by binding of CheY. In bacteria, the C-ring adopts the CW pose ~30% of the time due to stochastic interactions with CheY. We now better highlight that the CCW pose is the resting state and include an experimental structure of a CW-locked mutant, which shows a distinct pose.

4. Moreover, the manuscript did not provide any data to support the “CW” model.

→ Response: In the revised manuscript, we replaced the computational model with an experimental structure of a CW-locked mutant.

Decision Letter, first revision:

Message: 19th January 2024

Dear Tina,

Thank you for your patience while your manuscript "Structural basis for motor rotation and directional switching of bacterial flagella" was under peer-review at Nature Microbiology. Your revised manuscript has now been seen again by 2 of the original referees. You will see from their comments below that while they find your work of interest, one of the referees still feels that some important points have not been fully addressed. We are very interested in the possibility of publishing your study in Nature Microbiology, but would like to consider your response to these concerns in the form of a revised manuscript before we make a final decision on publication.

In particular, you will see that Referee #2 has significant concerns over the strength of the claim of 33mer stoichiometry over 34mer stoichiometry, given previously published literature, the use of the pKLR3 plasmid, and potential distribution in the data obtained. It would be essential that a revised manuscript tone down this claim, address the reviewer's concerns and discuss the potential limitations to the use of the pKLR3 construct and fairly

17discuss existing literature. In addition, the referee remained concerned about how well existing literature had been referenced and discussed in the revised manuscript. This point would also need to be addressed, alongside any other points in the report. To ensure that the study remains timely, we would also need these revisions to be completed very quickly and would ask you to resubmit within 1 week.

If you have not done so already please begin to revise your manuscript so that it conforms to our Article format instructions at <http://www.nature.com/nmicrobiol/info/final-submission/>

The usual length limit for a Nature Microbiology Article is six display items (figures or tables) and 3,000 words. We have some flexibility, and can allow a revised manuscript at 3,500 words, but please consider this a firm upper limit. There is a trade-off of ~250 words per display item, so if you need more space, you could move a Figure or Table to Supplementary Information.

Some reduction could be achieved by focusing any introductory material and moving it to the start of your opening 'bold' paragraph, whose function is to outline the background to your work, describe in a sentence your new observations, and explain your main conclusions. The discussion should also be limited. Methods should be described in a separate section following the discussion, we do not place a word limit on Methods.

Nature Microbiology titles should give a sense of the main new findings of a manuscript, and should not contain punctuation. Please keep in mind that we strongly discourage active verbs in titles, and that they should ideally fit within 90 characters each (including spaces).

Please include a data availability statement as a separate section after Methods but before references, under the heading "Data Availability". This section should inform readers about the availability of the data used to support the conclusions of your study. This information includes accession codes to public repositories (data banks for protein, DNA or RNA sequences, microarray, proteomics data etc...), references to source data published

alongside the paper, unique identifiers such as URLs to data repository entries, or data set DOIs, and any other statement about data availability. At a minimum, you should include the following statement: "The data that support the findings of this study are available from the corresponding author upon request", mentioning any restrictions on availability. If DOIs are provided, we also strongly encourage including these in the Reference list (authors, title, publisher (repository name), identifier, year). For more guidance on how to write this section please see: <http://www.nature.com/authors/policies/data/data-availability-statements-data-citations.pdf>

To improve the accessibility of your paper to readers from other research areas, please pay particular attention to the wording of the paper's opening bold paragraph, which serves both as an introduction and as a brief, non-technical summary in about 150 words. If, however, you require one or two extra sentences to explain your work clearly, please include them even if the paragraph is over-length as a result. The opening paragraph should not contain references. Because scientists from other sub-disciplines will be interested in your results and their implications, it is important to explain essential but specialised terms concisely. We suggest you show your summary paragraph to colleagues in other fields to uncover any problematic concepts.

If your paper is accepted for publication, we will edit your display items electronically so they conform to our house style and will reproduce clearly in print. If necessary, we will re-size figures to fit single or double column width. If your figures contain several parts, the parts should form a neat rectangle when assembled. Choosing the right electronic format at this stage will speed up the processing of your paper and give the best possible results in print. We would like the figures to be supplied as vector files - EPS, PDF, AI or postscript (PS) file formats (not raster or bitmap files), preferably generated with vector-graphics software (Adobe Illustrator for example). Please try to ensure that all figures are non-flattened and fully editable. All images should be at least 300 dpi resolution (when figures are scaled to approximately the size that they are to be printed at) and in RGB colour format. Please do not submit Jpeg or flattened TIFF files. Please see also 'Guidelines for Electronic Submission of Figures' at the end of this letter for further detail.

Figure legends must provide a brief description of the figure and the symbols used, within 350 words, including definitions of any error bars employed in the figures.

When submitting the revised version of your manuscript, please pay close attention to our [href="https://www.nature.com/nature-research/editorial-policies/image-integrity">Digital Image Integrity Guidelines](https://www.nature.com/nature-research/editorial-policies/image-integrity). and to the following points below:

- that unprocessed scans are clearly labelled and match the gels and western blots presented in figures.
- that control panels for gels and western blots are appropriately described as loading on sample processing controls

-- all images in the paper are checked for duplication of panels and for splicing of gel lanes.

Please include a statement before the acknowledgements naming the author to whom correspondence and requests for materials should be addressed.

Finally, we require authors to include a statement of their individual contributions to the paper -- such as experimental work, project planning, data analysis, etc. -- immediately after the acknowledgements. The statement should be short, and refer to authors by their initials. For details please see the Authorship section of our joint Editorial policies at http://www.nature.com/authors/editorial_policies/authorship.html

- * include a point-by-point response to any editorial suggestions and to our referees. Please include your response to the editorial suggestions in your cover letter, and please upload your response to the referees as a separate document.

- * ensure it complies with our format requirements for Letters as set out in our guide to authors at www.nature.com/nmicrobiol/info/gta/

- * state in a cover note the length of the text, methods and legends; the number of references; number and estimated final size of figures and tables

- *This url links to your confidential homepage and associated information about manuscripts you may have submitted or be reviewing for us. If you wish to forward this e-mail to co-authors, please delete this link to your homepage first.

Please ensure that all correspondence is marked with your Nature Microbiology reference number in the subject line.

Nature Microbiology is committed to improving transparency in authorship. As part of our efforts in this direction, we are now requesting that all authors identified as 'corresponding author' on published papers create and link their Open Researcher and Contributor Identifier (ORCID) with their account on the Manuscript Tracking System (MTS), prior to acceptance. This applies to primary research papers only. ORCID helps the scientific community achieve unambiguous attribution of all scholarly contributions. You can create and link your ORCID from the home page of the MTS by clicking on 'Modify my Springer

Nature account'. For more information please visit please visit www.springernature.com/orcid.

We hope to receive your revised paper within one week. If you cannot send it within this time, please let us know.

Yours sincerely,

Reviewer Expertise:

Referee #1:

Referee #2:

Referee #3:

Reviewers Comments:

Reviewer #1 (Remarks to the Author):

The authors made a great job in revising the manuscript. I am satisfied and congratulate them to this amazing piece of work.

Reviewer #2 (Remarks to the Author):

This resubmission of a previously-rejected manuscript includes substantial new data. Most importantly, the authors present an empirically-derived structure of the CW-locked flagellar rotor. A collateral result of this is an intriguing additional density in some data apparently binding to the top of FliG. The manuscript has also been substantially revised in face of previous comments comments.

Nevertheless, there are still a number of issues with the manuscript that would benefit from addressing to make this into the important publication that it is destined to become.

Claims of the MS-ring forming from 33 copies of FliF: Line 569, "However, during our data analysis, we performed extensive subset classification to identify whether other stoichiometries were present at lower abundance (Extended Data Fig. 1b). From this analysis, we are confident that the MS-ring is only a 33-mer when the C-ring is a 34-mer

under the conditions of our sample preparation.”, and Line 387: The claim “We exclusively observed 33-mer MS-rings in wild-type 34-mer C-rings” fails to capture what looks like a complicated image processing schema in Extended Data Fig. 1. The 2x5 table of different classes and corresponding percentages isn’t explained, but as far as I can tell is far from conclusive that the results are “exclusively ... 33-mer MS-rings”. The authors need to explain carefully what they did here. What of the other 42% of the data in C33? This is roughly half of the data with no evidence for stoichiometry of 33.

While I very much appreciated the speculation of the 4-degree tilt of the MS-ring to the C-ring as potentially being due to the FliG:FliG stoichiometric and symmetry mismatch, this section remains problematic. The authors respond to previous reviews, “In the revised version, we added a section to the discussion describing the existing controversy over MS-ring stoichiometry. We explicitly indicate that the MS-ring controversy is not resolved by our data. ”, and line 573, “From this analysis, we are confident that the MS-ring is only a 33-mer when the C-ring is a 34-mer under the conditions of our sample preparation.” – this is still uncharitable to the likely synthesis by Kawamoto et al of all of the work from many groups that native symmetry is very likely 34. If the authors have issue with this, they need to explicitly explain why. If not, they must acknowledge to non-specialist readers that the pKLR3 plasmid that they used lacks components that may template correct formation of FliF. Indeed, in the injectisome system, SctRST (=FliPQR) are needed for correct assembly of 24-fold symmetric SctDJ (=FliF); without SctRST, SctDJ assembles as 23-mers – see Butan et al., ‘High-Resolution View of the Type III Secretion Export Apparatus in Situ Reveals Membrane Remodeling and a Secretion Pathway’. It is very likely that the authors are seeing the same artefactual mis-assembly in the absence of the export gate template.

In the previous version of this manuscript I expressed my disappointment at poor referencing and real or perceived poor contemporary knowledge of the field by the authors. The manuscript has now been improved, but I see further disappointing errors (or lack of attention to detail), especially in the discussion of CW/CCW-switching:

- Line 493: “All past proposals” – which proposals? Where are the references? This proposal is recent, based on the speculative premise that MotA pentamers rotate. This phrasing obscures correct understanding of the field to non-specialists.
- Line 496: Reference 68 makes no reference to stator complex movement (the paper doesn’t even feature the word ‘clockwise’). Do the authors mean to cite Chang et al., ‘Molecular Mechanism for Rotational Switching of the Bacterial Flagellar Motor’? One mis-citation would be unfortunate, but so many is a pattern. Please be sure to properly acquaint yourselves with the literature and ensure due diligence in correctly citing work.
- The correct Chang et al paper (‘Molecular Mechanism for Rotational Switching of the Bacterial Flagellar Motor’) does not suggest a “a movement of MotA/B from the outer face of the C-ring to the inner face” – rather, this paper suggests a movement of FliG(C) to the outer rim of MotAB. Be clear and be accurate in your writing.

(The author’s rebuttal that “Among our authors are individuals with nearly 50 years of

experience in the flagellar motor and bacterial chemotaxis.” is yet greater reason to expect that you read and understand papers correctly!)

Author Rebuttal, first revision:

Reviewer #2 (Remarks to the Author):

This resubmission of a previously-rejected manuscript includes substantial new data. Most importantly, the authors present an empirically-derived structure of the CW-locked flagellar rotor. A collateral result of this is an intriguing additional density in some data apparently binding to the top of FliG. The manuscript has also been substantially revised in face of previous comments.

Nevertheless, there are still a number of issues with the manuscript that would benefit from addressing to make this into the important publication that it is destined to become.

Claims of the MS-ring forming from 33 copies of FliF: Line 569, “However, during our data analysis, we performed extensive subset classification to identify whether other stoichiometries were present at lower abundance (Extended Data Fig. 1b). From this analysis, we are confident that the MS-ring is only a 33-mer when the C-ring is a 34-mer under the conditions of our sample preparation.”, and Line 387: The claim “We exclusively observed 33-mer MS-rings in wild-type 34-mer C-rings” fails to capture what looks like a complicated image processing schema in Extended Data Fig. 1. The 2x5 table of different classes and corresponding percentages isn’t explained, but as far as I can tell is far from conclusive that the results are “exclusively ... 33-mer MS-rings”. The authors need to explain carefully what they did here. What of the other 42% of the data in C33? This is roughly half of the data with no evidence for stoichiometry of 33.

To improve the clarity, we added this information to the main text and greatly expanded the legend to ED Fig 1. We clarify that MS-rings could not be correctly classified when they were adjacent to a larger 34-fold symmetric structure. We then added more details on how particle subtraction was instrumental in distinguishing between C33 and C34. We also clarified that the MS-rings that could not be classified were associated with thin ice and were likely at the air-water interface. Given the reviewer’s interest, we performed a rigorous additional subset classification where we separated all particles that were not 33-mers (i.e. we isolated the 42%) and we imposed C32, C34, C35, and C36 symmetry on this subset. Even excluding all 33-mers, we could not identify other symmetries. Finally, we note that symmetry

23classification can be challenging and we cannot rule out that there are subpopulations that we cannot classify.

While I very much appreciated the speculation of the 4-degree tilt of the MS-ring to the C-ring as potentially being due to the FliG:FliG stoichiometric and symmetry mismatch, this section remains problematic. The authors respond to previous reviews, “In the revised version, we added a section to the discussion describing the existing controversy over MS-ring stoichiometry. We explicitly indicate that the MS-ring controversy is not resolved by our data. “, and line 573, “From this analysis, we are confident that the MS-ring is only a 33-mer when the C-ring is a 34-mer under the conditions of our sample preparation.” – this is still uncharitable to the likely synthesis by Kawamoto et al of all of the work from many groups that native symmetry is very likely 34. If the authors have issue with this, they need to explicitly explain why. If not, they must acknowledge to non-specialist readers that the pKLR3 plasmid that they used lacks components that may template correct formation of FliF. Indeed, in the injectisome system, SctRST (=FliPQR) are needed for correct assembly of 24-fold symmetric SctDJ (=FliF); without SctRST, SctDJ assembles as 23-mers – see Butan et al., ‘High-Resolution View of the Type III Secretion Export Apparatus in Situ Reveals Membrane Remodeling and a Secretion Pathway’. It is very likely that the authors are seeing the same artefactual mis-assembly in the absence of the export gate template.

→ R2 brings up important points about the templating machinery. The expression of isolated FliF in Kawamoto et al. also used plasmid-expressed FliF expressed in *E. coli*. We therefore suggest that the *E. coli* templating machinery (95% identical to Salmonella) was recruited in both cases. Thus, other conditions drove the stoichiometric differences.

Given the reviewer’s interest, we increased the text devoted to the discussion of the MS-ring stoichiometry. Throughout, we used language that included statements like “this is the symmetry that we observe under the conditions of our sample preparation” which acknowledges that sample preparation conditions do make a difference. The inserted text is below.

Lines 255 - 267 (results)

“The remaining MS-rings could not be classified into a stoichiometry. MS-rings that could not be classified correlated with grids that had thinner ice, suggesting that the MS-ring preferentially partitions at

the air-water interface. However, we cannot exclude the presence of other stoichiometries that we could not classify.

While the fold of each FliF subunit is generally consistent with previous reports^{5,9-11}, there are two notable differences. The first is the 33-mer stoichiometry. Some past structures show variable stoichiometry^{5,10}. Others suggest that the native stoichiometry is a 34-mer⁹...”

Lines 379 - 407 (Discussion):

Notably, there is currently no consensus over the stoichiometry for the MS-ring^{5,9,10,12}. Some cryoEM studies showed a range of stoichiometries^{5,10}. Other studies only identify a 34-mer⁹. The variable stoichiometry was interpreted as the MS-ring adapting to load. This largely leveraged parallels to the C-ring’s stoichiometry⁶⁷ and the number of bound stators⁶⁸, which can change in response to the strength of the attractant or load.

Studies showing only a 34-mer suggest that other stoichiometries arise from artifacts due to C-terminal proteolysis of FliF or incorrect templating during plasmid expression⁹. We can exclude C-terminal proteolysis affecting stoichiometry in our structure because we observe density for the full C-terminus of FliF bound to FliG (Fig. 2a, 3a, Extended Data Fig. 3a). In terms of templating, this 33-mer MS-ring and the previously published strict 34-mer⁹ were similarly expressed in *E. coli*. This suggests that the *E. coli* templating machinery can be recruited to assemble the Salmonella MS-ring and is also unlikely to underlie the stoichiometric difference. Nevertheless, we did not test conditions proposed to affect stoichiometry, which would be required to distinguish between an adaptive and a strict stoichiometry. For example, we did not coexpress the *S. enterica* templating machinery with the pKLR3 plasmid, and we did not grow cells under conditions with different attractants or different loads. Taken together, the origins of symmetry differences in MS-ring structures remain unclear at this time.

In the previous version of this manuscript I expressed my disappointment at poor referencing and real or perceived poor contemporary knowledge of the field by the authors. The manuscript has now been improved, but I see further disappointing errors (or lack of attention to detail), especially in the discussion of CW/CCW-switching:

- Line 493: “All past proposals” – which proposals? Where are the references? This proposal is recent, based on the speculative premise that MotA pentamers rotate. This phrasing obscures correct understanding of the field to non-specialists.

→ to improve the clarity, we removed the word “all” and moved the relevant references to be after the opening phrase of the sentence.

- Line 496: Reference 68 makes no reference to stator complex movement (the paper doesn't even feature the word 'clockwise'). Do the authors mean to cite Chang et al., 'Molecular Mechanism for Rotational Switching of the Bacterial Flagellar Motor'? One mis-citation would be unfortunate, but so many is a pattern. Please be sure to properly acquaint yourselves with the literature and ensure due diligence in correctly citing work.

→ We thank the reviewer for catching this. The citation is corrected.

- The correct Chang et al paper ('Molecular Mechanism for Rotational Switching of the Bacterial Flagellar Motor') does not suggest a “a movement of MotA/B from the outer face of the C-ring to the inner face” – rather, this paper suggests a movement of FliG(C) to the outer rim of MotAB. Be clear and be accurate in your writing.

→ We adjusted the language to indicate that the conformational change of the CW pose and the rotation of FliG_{D5} moves the MotA/B binding site to the inside of the ring. Our interpretation is that the cryoET is consistent with this proposal.

(The author's rebuttal that “Among our authors are individuals with nearly 50 years of experience in the flagellar motor and bacterial chemotaxis.” is yet greater reason to expect that you read and understand papers correctly!)

Decision Letter, second revision:

Message: Our ref: NMICROBIOL-23082163B

1st February 2024

Dear Tina,

Thank you for submitting your revised manuscript "Structural basis for rotation and

26directional switching of bacterial flagella" (NMICROBIOL-23082163B). We have looked into your revisions and pending minor revisions to comply with our editorial and formatting guidelines, we will be happy to publish your article in Nature Microbiology.

Please can you send us an unredacted word document version of the manuscript, in single column format by return email. We are about to do some detailed checks on the paper to provide you with a final checklist to make sure that everything fits with the formatting guidelines and to smooth the publication process. We will need the unredacted version before we go forwards with these checks. Once we have this, we hope to get a final checklist to you within a week.

Please do not upload any other final materials or make any other revisions until you receive this additional information from us.

Thank you again for your interest in Nature Microbiology Please do not hesitate to contact me if you have any questions.

Sincerely,

Final Decision Letter:

Message 12th March 2024

:

Dear Tina,

I am pleased to accept your Article "CryoEM structures reveal how the bacterial flagellum rotates and switches direction" for publication in Nature Microbiology. Thank you for having chosen to submit your work to us and many congratulations.

You may wish to make your media relations office aware of your accepted publication, in case they consider it appropriate to organize some internal or external publicity. Once your paper has been scheduled you will receive an email confirming the publication details. This is normally 3-4 working days in advance of publication. If you need additional notice of the date and time of publication, please let the production team know when you receive the proof of your article to ensure there is sufficient time to coordinate. Further information on

27our embargo policies can be found here:
<https://www.nature.com/authors/policies/embargo.html>

Please note that *Nature Microbiology* is a Transformative Journal (TJ). Authors may publish their research with us through the traditional subscription access route or make their paper immediately open access through payment of an article-processing charge (APC). Authors will not be required to make a final decision about access to their article until it has been accepted. Find out more about Transformative Journals

We welcome the submission of potential cover material (including a short caption of around 40 words) related to your manuscript; suggestions should be sent to Nature

Microbiology as electronic files (the image should be 300 dpi at 210 x 297 mm in either TIFF or JPEG format). Please note that such pictures should be selected more for their aesthetic appeal than for their scientific content, and that colour images work better than black and white or grayscale images. Please do not try to design a cover with the Nature Microbiology logo etc., and please do not submit composites of images related to your work. I am sure you will understand that we cannot make any promise as to whether any of your suggestions might be selected for the cover of the journal.

With kind regards,